# MoESD: Unveil Speculative Decoding's Potential for Accelerating Sparse MoE

**Zongle Huang**[1,3]  **Lei Zhu**[2]  **Zongyuan Zhan**[2]  **Ting Hu**[2]  **Weikai Mao**[2]  **Xianzhi Yu**[2]
**Yongpan Liu**[1,3†]  **Tianyu Zhang**[2†]
[1]Tsinghua University  [2]Huawei Noah's Ark Lab  [3]BNRist
{huangzl23}@mails.tsinghua.edu.cn  {ypliu}@tsinghua.edu.cn
{zhulei168,zhanzongyuan,huting35,maoweikai,yuxianzhi,zhangtianyu59}@huawei.com

## Abstract

Large Language Models (LLMs) have achieved remarkable success across many applications, with Mixture of Experts (MoE) models demonstrating great potential. Compared to traditional dense models, MoEs achieve better performance with less computation. Speculative decoding (SD) is a widely used technique to accelerate LLM inference without accuracy loss, but it has been considered efficient only for dense models. In this work, we first demonstrate that, under medium batch sizes, MoE surprisingly benefits more from SD than dense models. Furthermore, as MoE becomes sparser – the prevailing trend in MoE designs – the batch size range where SD acceleration is expected to be effective becomes broader. To quantitatively understand tradeoffs involved in SD, we develop a reliable modeling based on theoretical analyses. While current SD research primarily focuses on improving acceptance rates of algorithms, changes in workload and model architecture can still lead to degraded SD acceleration even with high acceptance rates. To address this limitation, we introduce a new metric *target efficiency* that characterizes these effects, thus helping researchers identify system bottlenecks and understand SD acceleration more comprehensively. For scenarios like private serving, this work unveils a new perspective to speed up MoE inference, where existing solutions struggle. Experiments on different GPUs show up to 2.29x speedup for Qwen2-57B-A14B at medium batch sizes and validate our theoretical predictions.

## 1   Introduction

Recent years have witnessed remarkable success in Large Language Models (LLMs), with Mixture of Experts (MoE) architectures showing tremendous potential. Unlike dense models use a single feed-forward network (FFN) to process all inputs, MoE models replace the FFN with multiple specialized "expert" networks plus a router that selectively activates only a few experts for each input token. Such sparsity in structure enables MoEs with more parameters to achieve higher computational efficiency, and multiple state-of-the-art LLMs, such as DeepseekV3 [1] and Qwen2.5-Max [2], are all MoEs. MoE model architectures are evolving toward larger scales with increased sparsity [3, 4, 1] and more balanced workload distribution among experts [5, 6].

Speculative decoding (SD) is a lossless technique to accelerate LLM inference, but conventional wisdom suggests that its efficacy diminishes when applied to MoEs. In SD, a smaller draft model is introduced to rapidly generate multiple candidate tokens, while the larger target model verifies these predictions in parallel, preserving only correctly speculated tokens. For dense models' inference, the time taken to generate a single token and verify multiple ones is roughly the same, as both tasks require the full set of parameters to be loaded once. Therefore, SD gains acceleration through fewer forward

---

† Corresponding Author.

rounds of the target model and shorter decoding time of the draft model. However, this acceleration has been demonstrated to diminish in MoEs [7, 8], as the multiple draft tokens in verification activate more experts than a single token, leading to larger memory access and significantly longer verification time compared to a standard decoding step.

In this work, we challenge the conventional belief and demonstrate that, under a moderate batch size, SD can be more effective for MoEs than for dense models. Our key insight is that when the batch size is moderate such that all experts are already activated in a single decoding step, verifying multiple draft tokens will not incur additional expert parameter loading costs. Furthermore, as the MoE becomes sparser, each expert processes fewer tokens per parameter loading, leading to lower utilization of arithmetic units and thereby creating greater acceleration opportunities for SD.

The insight above is supported by comprehensive theoretical analyses, through which we identify a new metric *target efficiency* to quantify how systemic factors (such as workload and target model architecture) affect SD speedup. In contrast to existing SD works that use acceptance rate [9, 10, 7, 11], an algorithmic metric to evaluate how accurately the draft model speculates the target model, our proposed target efficiency isolates extrinsic factors like algorithm selection and focuses on intrinsic system bottlenecks caused by the target model's computational and memory access requirements. As demonstrated in the following sections, even with similar acceptance rates, systemic factors can greatly impact SD effectiveness, making our metric target efficiency necessary for a comprehensive understanding of SD acceleration.

As a further step, we build a quantitative modeling of SD speedup for MoE based on these theoretical analyses. The consistent matching between our modeling and experiment results confirms the reliability of our analyses. Additionally, the modeling itself provides an approach for analyzing the execution time of different components, making the end-to-end SD acceleration results more transparent and explainable.

Our work offers a new perspective for lossless MoE acceleration, particularly well-suited for private serving scenarios [12, 13, 14]. Private serving has gained popularity among enterprises seeking to safeguard data and model security, with typical applications such as in-house chatbots. These environments typically process moderate batches containing tens of requests. Additionally, our findings can be applied to latency-critical scenarios where large batch sizes are infeasible, or memory-restricted environments where MoEs exceed GPU capacity.

In Summary, the main contributions of our work are:

- We refine the conventional belief that speculative decoding cannot effectively accelerate MoEs, demonstrating that under moderate batch sizes, SD is actually more effective in a wider range of batch sizes for sparser MoEs than dense models.

- Based on theoretical analysis, we developed a reliable modeling for SD speedup, thus making the acceleration process transparent and explainable. Existing metrics only assess algorithmic optimization efficiency and cannot fully explain SD speedup, so we introduce a new systemic metric *target efficiency* that reveals speedup opportunities inherent in the target model.

- Our findings can be applied to accelerate scenarios like private serving. Experiments on various GPUs with the Qwen2-57B-14A-Instruct model demonstrate that SD achieves the highest speedup at the moderate batch size, reaching 2.29x. These experiments also validate our theoretical prediction that SD is more favorable for sparser MoEs.

## 2   Related Work

**MoE acceleration.**   MoE has emerged as a promising LLM architecture, and many techniques optimize its inference. Model compression methods, including pruning [15, 16], quantization [17, 18], distillation [19, 20], and decomposition [21, 22], have been applied to MoEs and achieved great acceleration. They sacrifice model quality for speedup, as in dense models. When MoEs are too large to fit in GPU memories and offloading becomes a necessity, several system-level approaches have emerged to optimize inference latency through improved scheduling techniques. Expert prefetching [23, 24] predicts and pre-loads experts for upcoming layers based on previous activation patterns, thus overlapping expert loading with current layer computation. Expert caching [25, 26] caches most frequently activated experts in GPU memory, leveraging expert locality to reduce expensive

offloading. Compared to them, our work unveils a new perspective for MoE acceleration that is lossless and doesn't depend on expert imbalance.

**Speculative Decoding.** Speculative decoding (SD), initially proposed by [10] and [9], has emerged as a widely adopted technique for accelerating LLM inference without sacrificing generation quality. Basic SD employs a smaller model to rapidly generate draft tokens, which are then verified in parallel by the target model that needs to be accelerated. Afterwards, more algorithms are developed to lift the acceptance rate of draft tokens. [27, 11, 28, 29, 7, 30, 31] adopt tree-structured generation patterns rather than chains to explore a broader range of potential completions. [1, 11, 7, 30, 31] propose to replace draft models with specifically trained speculative heads integrated in the target model.

Despite advances in SD algorithms, it has long been considered ineffective for *large batches* [32, 33, 27] or *MoE* [7, 8], since the verification time in these cases significantly increases. Until recently, MagicDec [34] first challenged that in long-sequence regimes, SD can effectively accelerate *large batches*, primarily due to the significantly increased KV cache altering the computation-to-memory access ratio of the model. However, SD research for *MoE* remains unexplored. In response, our work fills this gap, unveiling that under certain conditions, SD can effectively accelerate MoE models.

# 3 Theoretical Analysis

In this section, we present the theoretical analyses supporting our conclusion that SD can be more effective for MoE than dense models at moderate batch sizes. We begin by formalizing general SD speedup and introducing our new metric *target efficiency* (Sec. 3.1). Then, we focus on MoEs, analyzing how workload and MoE sparsity collectively affect the number of activated experts and SD speedup (Sec. 3.2). Based on these analyses, we develop a performance model that aligns with GPU results (Sec. 3.3). We further discuss the practical value of our theoretical findings (Sec. 3.4).

**Preliminaries.** LLM inference time is collaboratively determined by computation and memory access. When an operator is processed on a GPU, memory access and computation operations are pipelined and overlapped, causing the more time-consuming operation to become the bottleneck and determine the overall processing time, as depicted by the roofline model [35, 36]. The roofline model ridge point (RP) of hardware and the arithmetic intensity (AI) of software are defined as Eq. 1. When AI < RP, the system is *memory-bound*, and adding more computation will not significantly increase processing time. When AI > RP, the system is *compute-bound*, and increases in computation will directly reflect in processing time. In this paper, when we describe a system as "more memory-bound", we mean $\frac{\text{AI}}{\text{RP}}$ is smaller.

$$\mathbf{RP} = \frac{\text{peak computation power } \textit{(unit: Flops)}}{\text{peak memory bandwidth } \textit{(unit: bytes/second)}} \quad \mathbf{AI} = \frac{\text{computation operation } \textit{(unit: times)}}{\text{memory access volume } \textit{(unit: bytes)}} \quad (1)$$

## 3.1 Formulation of Speculative Decoding Speedup and Target Efficiency

We first formalize the processing time of speculative decoding, denoted as $T_{SD}$. To generate a sequence of length $S$, SD goes through $R$ rounds, each containing three stages: ① the draft model proposes $\gamma$ tokens as specified by the speculation strategy; ② the target model verifies these tokens; ③ rejection sampling [9] discards incorrectly predicted tokens based on logits from target and draft models. We use $T_T(b, s)$ and $T_D(b, s)$ to represent the time for once forwarding of the target and draft model, respectively, where $b$ and $s$ are the formal arguments for batch size and the number of tokens to process. Therefore, the time for processing a batch containing $B$ requests is given by:

$$T_{SD} = R \times (T_{propose} + T_{verify} + T_{reject}) = R \times \left( \gamma \cdot T_D(B, 1) + T_T(B, \gamma) + T_{reject} \right) \quad (2)$$

Then the speedup of SD to normal auto-regression decoding $T_{AR}$ is given by:

---

Since we work with typical sequence lengths and moderate batch sizes, the impact of KV-cache on performance is limited, allowing us to omit the already generated sequence length from our analysis. For cases where KV-cache becomes the dominant factor, see [34].

$$Speedup = \frac{T_{AR}}{T_{SD}} = \frac{S \cdot T_T(B,1)}{R \cdot \left( \gamma \cdot T_D(B,1) + T_T(B,\gamma) + T_{reject} \right)} \tag{3}$$

$$= \frac{S}{R} \cdot \frac{1}{\gamma \cdot \frac{T_D(B,1)}{T_T(B,1)} + \frac{T_T(B,\gamma)}{T_T(B,1)} + \frac{T_{reject}}{T_T(B,1)}} \tag{4}$$

$\frac{S}{R}$ represents the average length of accepted tokens per SD round, which can be further expressed as $\sigma \times (\gamma + 1)$. Here, $\sigma$ is the ratio of actually generated tokens to the theoretical maximum if all draft tokens were accepted. We note that $\sigma$ differs from the acceptance rate $\alpha$ commonly referenced in previous works [10, 9, 7], which represents the probability of the target model accepting a new draft token given the prefix. $\sigma$ can be computed from $\alpha$ as shown in Eq. 5. The numerator follows from [10], and the denominator accounts for all $\gamma$ draft tokens accepted, plus a bonus token generated during the forward verification pass.

$$\sigma = \frac{expected\ generated\ tokens}{maximal\ possible\ accepted\ tokens} = \frac{\frac{1-\alpha^{\gamma+1}}{1-\alpha}}{\gamma + 1} \tag{5}$$

The denominator of Eq. 4 consists of three terms. $\frac{T_D(B,1)}{T_T(B,1)}$ is the ratio of draft-model forward time over target-model forward time, reflecting the relative volume of draft and target models. It is also kept small (usually less than 1/10 [27, 7, 10]) to ensure the speculation is efficient. $\frac{T_{reject}}{T_T(B,1)}$ is even smaller, since $T_{reject}$ only involves sampling rather than model inference. $\frac{T_T(B,\gamma)}{T_T(B,1)}$, which is the ratio of multi-token forward time over single-token forward time, has the biggest value among these three items and significantly affects the final speedup. As indicated by Eq. 4, its increase causes speedup reduction. Two different factors drive its increase, explaining SD's ineffectiveness under (1) *large batches* for both dense models and MoE, and (2) *MoE* with small batches, respectively:

(1) The compute-boundness. The model's $\frac{T_T(B,\gamma)}{T_T(B,1)}$ approaches 1 when more memory-bound (smaller batch size $B$) but increases to $\gamma$ when more compute-bound (larger batch size $B$).

(2) The extra memory loads. For small $B$s, $T_T(B,\gamma)$ is notably greater than $T_T(B,1)$ as more experts are activated and need to be loaded. Since the system is still memory-bound at small $B$s, memory load profoundly determines the processing time.

Therefore, we define **target efficiency** as $\frac{T_T(B,1)}{T_T(B,\gamma)}$, which helps understand the systemic causes of SD acceleration degradation as listed above. Our experiments shown in Fig. 2 demonstrate that target efficiency consistently reflects the trend of SD speedup variations. Despite the importance of this value, previous works have rarely noticed it, primarily due to differences in research focus. Previous SD research mainly addresses the question by lifting the acceptance rate:

*Given the target model, which **draft model or algorithm** achieve better speedups?*

In contrast, our work focuses on the following question by examining target efficiency:

*Under the same level of algorithmic optimization, which types of **target models or workloads** are more favorable for SD?*

We believe target efficiency help researchers understand SD more comprehensively. Even when target-draft pairs have the same acceptance rate $\alpha$, changes in the target model's architecture and the workload can significantly affect overall speedup. By introducing target efficiency, we can decouple algorithmic optimization from systemic optimization, thus helping to identify the systemic acceleration bottlenecks and assess potential speedup.

### 3.2 Moderate Batch Size Enables Speculative Decoding Speedup for MoE

Although SD is ineffective for MoE under small batches, we demonstrate in this subsection that at moderate batch sizes—an overlooked regime in previous studies—SD speedup increases and benefits more from MoE with higher sparsity. Essentially, when the batch size falls within ranges where

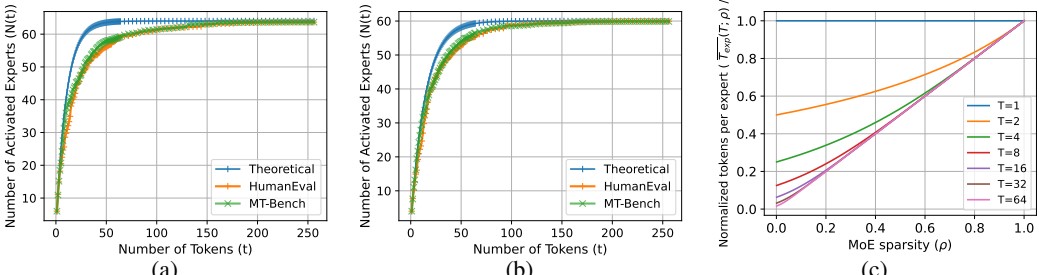

Figure 1: Activation status and workload of experts. (a) and (b): Comparison between theoretical and actual number of activated experts $N(t)$ on different datasets. (a) is for Deepseek-V2-Lite-Chat ($\rho = 6/62$) and (b) is for Qwen1.5-MoE-Chat ($\rho = 4/60$). (c): Normalized number of tokens to process per expert ($\overline{T_{exp}}$) versus MoE sparsity ($\rho$) for given input token count $T$.

all experts are activated but remain far from being assigned adequate workloads, FFNs become memory-bound, presenting an opportunity to leverage computational power almost for free through SD. To demonstrate this, we first formalize the expected number of activated experts, and then show MoE FFNs become more memory-bound as the model becomes sparser.

We use the Bernoulli random variable $X$ to indicate the activation for experts: $X_i = 1$ for expert $i$ being activated, $0$ otherwise. For simplicity, we assume $X$s are i.i.d. Then, the expected number of activated experts $N$ can be expressed as Eq. 6, where $E$ denotes the total expert count and $Pr(X_i)$ represents the probability that the $i^{th}$ expert is activated.

$$N = \sum_i \mathbb{E}[X_i] = \sum_i Pr(X_i) = E \cdot Pr(X) \tag{6}$$

Given $t$ tokens passed through the MoE gate, then $Pr(X)$ is expressed as Eq. 7. $K$ denotes the number of activated experts per token, which is an architectural hyperparameter for MoE:

$$Pr(X) = 1 - Pr(\text{ None of the } t \text{ tokens activates the expert }) = 1 - (\frac{E-K}{E})^t \tag{7}$$

Therefore, the overall expression of $N(t)$ is given by Eq. 8. Our derivation assumes uniformly activated experts, which is reasonable for well-trained MoE models. Load imbalance among experts can lead to routing collapse [5] and decrease computational efficiency in expert-parallel deployment [6], so state-of-the-art MoE models are typically trained with methods like incorporating auxiliary loss [37, 6] to ensure that experts have balanced loads. The experiment results also verified our theoretical derivation of $N(t)$, as shown in Fig. 1a and 1b.

$$N(t) = E \cdot \left(1 - (\frac{E-K}{E})^t\right) \tag{8}$$

We then solve how many tokens can lead to full activation. Since $N(t)$ asymptotically approaches $E$ when $t$ tends to infinity, and in practice $N(t)$ should be a finite integer, we deem $N(t) > \tau E$ as almost full activation, where $\tau$ is usually a large ratio such as 0.95. We further express $K = \rho E$, where $\rho$ is the sparsity of MoE, then the token threshold $T_{thres}$ can be solve by:

$$N(T_{thres}) = E \cdot \left(1 - (1-\rho)^{T_{thres}}\right) \geq \tau E \quad \Rightarrow \quad T_{thres} = \lceil \log_{(1-\rho)}(1-\tau) \rceil \tag{9}$$

Therefore, when $B$ exceeds $T_{thres}$, the number of activated experts saturates, causing the $B\gamma$ tokens in verification to incur only marginally larger memory access. Having addressed the second factor (namely, extra memory loads) for $\frac{T_T(B,\gamma)}{T_T(B,1)}$'s increase analyzed in Sec. 3.1, we now turn to the potential limitations caused by the first factor of compute-boundness. If such $B$s make the system compute-bound, SD would also fail to accelerate MoE effectively. Our answer to this concern is: Sparser MoEs *delay* the transition from memory-bound to compute-bound when input tokens count increases.

We have obtained that given $t$ tokens, $N(t)$ experts are activated. Since each token activates $K$ experts, the number of tokens each expert needs to process on average $\overline{T_{exp}}$ can be computed as:

$$\overline{T_{exp}}(t;\rho) = \frac{t \cdot K}{N} = \frac{t \cdot (\rho E)}{E \cdot \left(1 - (1-\rho)^t\right)} = \frac{\rho t}{1 - (1-\rho)^t} \tag{10}$$

---

**Algorithm 1** The Modeling of SD Speedup and Corresponding Fitting Method

---

1: **Measurement Input**: A total of $m$ measurements denoted as $\mathbf{M}$. Each $\mathbf{M}_i, i = 1, 2, ..., m$ contains the attributes including batch size $B$, draft length $\gamma$, number of activated experts per token $K$, total number of experts $E$, the ratio of accepted token counts to the maximal possible accepted tokens $\sigma$, *Speedup* for the actual speedup achieved.

2: **Output**: The optimal fitting parameter *params\**.

3: **def ComputeSpeedup**(*params*, $B$, $\gamma$, $K$, $E$, $\sigma$):         ▷ Compute the SD Speedup

4:  *bias*, $k_1$, $k_2$, $k_3$, *draft_bias*, *draft_k*, *reject_bias*, *reject_k*, $\lambda$, $s$ = *params*  ▷ Unpack parameters

5:  $N_{ar} = E \cdot (1 - ((E - K)/E)^B)$,   $T_{ar} = B \cdot K/N_{ar}$    ▷ Compute AR forward time

6:  $ar\_time = bias + k_1 \cdot G(B; \lambda RP, s) + k_2 \cdot N_{ar} + k_3 \cdot G(T_{ar}; \lambda RP, s)$

7:  $N_{sd} = E \cdot (1 - ((E - K)/E)^{B\gamma})$,   $T_{sd} = B \cdot \gamma \cdot K/N_{sd}$   ▷ Compute SD forward time

8:  $verify\_time = bias + k_1 \cdot G(B\gamma; \lambda RP, s) + k_2 \cdot N_{sd} + k_3 \cdot G(T_{sd}; \lambda RP, s)$

9:  $draft\_time = draft\_bias + draft\_k \cdot G(B; \lambda RP, s)$   ▷ Compute draft model forward time

10:  $reject\_time = reject\_bias + reject\_k \cdot B$     ▷ Compute rejection sampling time

11:  $Speedup = \sigma \cdot (\gamma + 1) \cdot \frac{ar\_time}{draft\_time + ar\_time + verify\_time + reject\_time}$  ▷ Compute the speedup as formalized in Eq. 4

12:  return *Speedup*

13: $params\text{*} = \underset{params}{\operatorname{argmin}} \frac{1}{2} \sum_{i=1}^{m} \Big( \textbf{\textit{ComputeSpeedup}}(params, \mathbf{M}_i.B, \mathbf{M}_i.\gamma, \mathbf{M}_i.K, \mathbf{M}_i.E, \mathbf{M}_i.\sigma) - \mathbf{M}_i.Speedup \Big)^2$

  ▷ Decide the optimal *params\** by fitting the model to the measured inputs using the least squares criterion.

---

As proven in Appendix and shown in Figure 1c, given $t = T > 1$, $\overline{T_{exp}}(T; \rho)$ decreases with $\rho$, indicating that as MoE becomes sparser, each expert processes *fewer tokens* per parameter loading. Consequently, the system running sparser MoEs is more *memory-bound*, leading to lower utilization of arithmetic units. The verification stage can therefore leverage these spare resources without notably increasing processing time. In contrast, dense models are extreme cases with $\rho = 1$, where the FFN consistently approaches the maximal possible arithmetic intensity of $T$, and the system transitions rapidly to the compute-bound regime as $T$ increases.

We should note that our conclusion is based on a relatively large MoE FFN portion in the whole model, which holds true for current MoE models whose most parameters are experts. In a hypothetical extreme case where Attention dominates and the MoE FFN is negligible, MoE's sparsity would have only a limited impact on overall system performance as indicated by Amdahl's Law.

### 3.3   A Modeling Method for Speculative Decoding Speedup

Given the numerous factors affecting final speedup, quantitatively understanding each factor's impact is challenging. Therefore, we developed a modeling method that makes SD speedup results more *explainable* and *transparent*. As demonstrated by Eq. 4, the core of modeling SD speedup lies in characterizing the model's forward pass time. Based on theoretical analysis in previous sections, we identified three primary factors affecting forward execution: (1) the roofline model effect, (2) the number of active experts, and (3) expert load. Since GPU execution is dynamic in practice, and not all operators are optimized to their theoretical limits, we introduced several parameters for relaxation. The values of these parameters are then automatically determined by fitting GPU measurements. These factors and their impacts on execution time are examined as follows.

**(1) The roofline model effect.**   It manifests as execution time increases with token counts $t$, with a growth *rate* that starts slow, then accelerates, and finally stabilizes. The underlying reasons are as follows. When $t$ is small, parameter loading time exceeds computation time, creating a memory bottleneck. Therefore, given the parameter volume, the memory access time is stable (memory-bound regime). As $t$ increases, computation time exceeds parameter loading time and becomes the bottleneck. With fixed arithmetic units in the hardware, computation time scales linearly with computational load (compute-bound regime). To characterize this trend, we design $G(t; \lambda RP, s)$ as shown in Eq. 11, where $\lambda RP$ represents the transition point between regimes, and $s$ controls the increasing rate of execution time. Here, $RP$ follows Eq. 1, and $\lambda$ is a constant less than 1 that accounts for practical limitations in memory bandwidth utilization. $G(t)$ exhibits a gradually increasing slope before the transition point, then shifts to a linear function afterwards, maintaining first-order gradient

continuity at the transition.

$$G(t; \lambda RP, s) = \begin{cases} s^t, & t \leq \lambda RP \\ s^{\lambda RP} + \left(\frac{\mathbf{d}(s^t)}{\mathbf{d}t}\big|_{t=\lambda RP}\right)(t - \lambda RP) = s^{\lambda RP}\left(1 + ln(s) \cdot (t - \lambda RP)\right), & t > \lambda RP \end{cases}$$
$$(11)$$

**(2) The number of activated experts.** When it increases, the memory access volume increases, thus adding to the final processing time. We use the derived Eq. 8 of $N$ to characterize how workload and model architecture affects the number of activated experts.

**(3) Expert load.** This refers to the fact that after token distribution through the MoE gating, each expert processes only a subset of tokens $\overline{T_{exp}}(t; \rho)$ rather than the entire input token count $t$. Therefore, we should use $G(\overline{T_{exp}})$ rather than $G(t)$ when applying the roofline model to MoE experts. This corroborates our theoretical conclusion that sparser MoEs *delay* the transition from memory-bound to compute-bound when input tokens count increases.

For the MoE target model, factors (1), (2), and (3) are all involved. We combine them in a first-order style and introduce parameters *bias*, $k_1$, $k_2$, and $k_3$ to adjust for non-ideal factors in actual GPU execution, with the full expression shown in lines 6 and 8 of Alg. 1. These parameters have clear practical meanings: *bias* represents the time required to load fixed parameters; $k_2 \cdot N$ represents the time needed to load $N$ activated experts; $k_1 \cdot G(t)$ and $k_3 \cdot G(\overline{T_{exp}})$ describe the *incremental* trend in execution time as the number of tokens increases. For the draft model, only factor (1) is involved since it is usually dense, with the modeling form shown in line 9 of Alg. 1.

With the expression of SD speedup determined, we fit the measurement inputs to automatically determine the relaxation parameter values, with the optimization criterion being the minimization of Mean Squared Error (MSE) between the model outputs and the ground truth, as shown in line 13 of Alg. 1. By applying these optimized parameters in our model (i.e., the *ComputeSpeedup* function in line 3), we obtain the complete modeling for SD speedup. An illustrative diagram of this process and more fitting details are provided in Appendix C.

Since our theoretical analyses capture the primary tradeoffs and provide a solid foundation for the modeling, the fitting is very efficient. The fitting results with 21 measurements are displayed in Fig. 4, which show consistent trends with GPU results under various cases. These results validate the reliability of our modeling, thereby establishing it as an effective tool for analyzing the components of the model's forward pass and quantitatively understanding the tradeoffs between different factors. As shown in Sec. 4.2, we explain some unexpected results with the help of the model.

### 3.4 Practical Values of Theoretical Findings

While previous sections focuses on theoretical analysis, this section demonstrates how these findings translate to practical speedups. Our theoretical analysis has already revealed that SD speedup for MoE is most effective at *moderate* batch sizes, with its trend initially increasing and then decreasing. We discuss their practical values considering both basic deployment and extended configurations.

**Basic deployment.** (1) Moderate batch sizes are common in private serving, which are increasingly adopted for data security, with representative applications like enterprise in-house chatbots. (2) When latency requirements are strict, large batch sizes are often not feasible. LLM serving must satisfy multiple service level objectives (SLOs) [38], including time-to-first-token (TTFT) and time-per-output-token (TPOT). Large batches reduce per-request computational resources, causing latency violations. In such cases, moderate batch sizes are common. (3) Our work actually reveals that SD on MoE relaxes the traditional *latency-throughput trade-off*. Specifically, MoE models exhibit a regime where SD speedup increases (lower latency) alongside larger batch sizes (higher throughput).

From the model's perspective, moderate batch sizes represent an "*efficiency gap*" in MoEs. At this scale, all parameters must be loaded (unlike small batches with selective expert activation), yet GPU FLOPs are not fully utilized (unlike large batches). Our findings provide a novel perspective to address this efficiency challenge without compromising model quality.

**Extended configurations.** We consider typical system optimizations on MoE like *offloading* and *expert parallelism (EP)*. When MoE models exceed GPU memory capacity, FFN parameters are offloaded to CPU memory [39]. This degrades parameter loading bandwidth from GPU memory bandwidth to much lower PCIe bandwidth, making the system more memory-bound. Consequently,

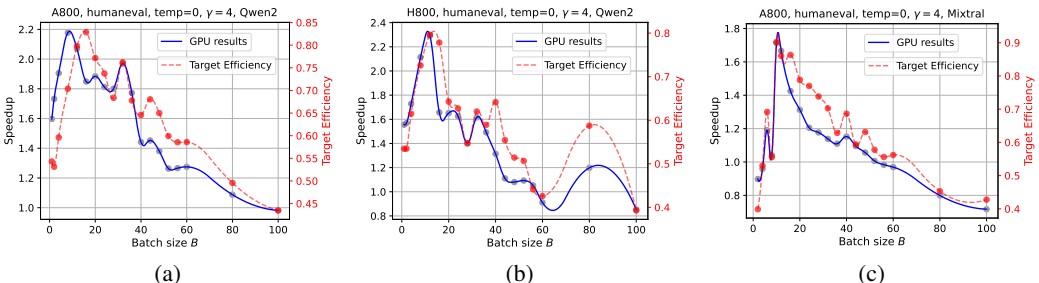

(a)               (b)               (c)

Figure 2: SD speedup (left y-axis) as a function of batch size and corresponding target efficiency values (right y-axis). Across different hardware platforms and MoE models, SD speedup first increases and then decreases, verifying our theoretical predictions. The target efficiency shows consistent trends with final speedup, validating its effectiveness.

Table 1: The peak speedup ($\mathbf{x}$) of SD across different datasets, temperatures, $\gamma$s and models on 2xA800

| Device | Dataset | Temp | $\gamma = 2$ | | | | $\gamma = 3$ | | | | $\gamma = 4$ | | | |
|---|---|---|---|---|---|---|---|---|---|---|---|---|---|---|
| | | | $T_{AR}$ | $T_{SD}$ | $\sigma$ | $\mathbf{x}$ | $T_{AR}$ | $T_{SD}$ | $\sigma$ | $\mathbf{x}$ | $T_{AR}$ | $T_{SD}$ | $\sigma$ | $\mathbf{x}$ |
| Qwen2 | humaneval | 0.0 | 18.89 | 11.61 | 0.94 | 1.63 | 15.93 | 8.11 | 0.93 | 1.96 | 15.93 | 7.31 | 0.91 | **2.18** |
| | humaneval | 1.0 | 19.13 | 12.93 | 0.83 | 1.48 | 21.20 | 14.09 | 0.73 | 1.50 | 19.13 | 11.14 | 0.67 | **1.72** |
| | mtbench | 0.0 | 20.92 | 16.70 | 0.71 | 1.25 | 16.00 | 12.43 | 0.62 | **1.29** | 20.92 | 17.53 | 0.55 | 1.19 |
| | mtbench | 1.0 | 21.15 | 17.33 | 0.68 | 1.22 | 19.09 | 14.83 | 0.57 | **1.29** | 19.09 | 15.93 | 0.48 | 1.20 |
| Mixtral | humaneval | 0.0 | 20.86 | 12.47 | 0.78 | 1.67 | 21.00 | 12.46 | 0.66 | 1.69 | 20.86 | 11.69 | 0.58 | **1.79** |
| | humaneval | 1.0 | 21.52 | 15.58 | 0.61 | **1.38** | 21.39 | 16.03 | 0.46 | 1.33 | 21.48 | 16.23 | 0.39 | 1.32 |
| | mtbench | 0.0 | 21.61 | 16.10 | 0.61 | **1.34** | 21.61 | 16.43 | 0.46 | 1.32 | 21.36 | 16.89 | 0.39 | 1.26 |
| | mtbench | 1.0 | 21.33 | 17.70 | 0.53 | **1.21** | 21.33 | 17.84 | 0.43 | 1.20 | 21.33 | 18.05 | 0.35 | 1.18 |

additional computation does not significantly increase processing time, creating favorable conditions for SD. Notably, existing optimizations like expert prefetching [23, 24] and caching [25, 26] lose efficiency under moderate batch sizes since nearly all experts are activated.

Our findings are also compatible with EP. In EP, experts are distributed across multiple GPUs, which affects neither $N(t)$ nor $\overline{T_{exp}}$, making our previous analyses remain valid. Since components besides MoE FFN are also parallelized, MoE FFN continues to constitute a significant portion of processing time, allowing memory-boundedness effects to remain observable in end-to-end performance. Notably, under extensive EP configurations, the inefficiency of SD for MoE at a small batch size may vanish, considering the additional memory bandwidth offered by large amounts of EP GPUs.

## 4 Experiments

**Models and datasets.** We evaluated two MoE configurations with SD: Qwen2-57B-A14B-Instruct with Qwen2-0.5B-Instruct [40], representing same-family drafting, and Mixtral-8x7B-Instruct-v0.1 [41] with Eagle [7], representing specialized speculation heads. When we need to examine MoEs with different sparsity, we modify the `num_experts_per_token` in the model's config.json file. For comparison with dense models, we use Opt-30b and Opt-350m [42] as the target and draft models. Models are evaluated on HumanEval [43] and MT-bench [44] datasets for code generation and conversation tasks, following previous works [7, 45, 11]. The tokenized prompt lengths range from 38 to 391 tokens for HumanEval and 5 to 356 tokens for MT-bench.

**Frameworks and hardware.** We used the existing vllm [46] framework for our experiments to verify theoretical predictions. Vllm supports batched speculative decoding, cudagraph optimization, and reports comprehensive data such as $T_D, T_T, T_{reject}$ and $\sigma$, thus being suitable for our experiments. To prevent unstable performance at the beginning, all data were obtained by averaging the results from the last five of the total ten runs. We conducted experiments on different hardware platforms including 2xA800, 2xH800, 4xA800, 4xL40.

### 4.1 Speedup Trend of Speculative Decoding for MoE

Figure 2 plots the end-to-end SD speedup (left y-axis) for MoE across various settings, validating our theoretical prediction about acceleration behavior. As batch size grows, speedup initially increases

Table 2: The peak speedup (x) of SD across different datasets, temperatures, γs and hardware on Qwen2

| Device | Dataset | Temp | $\gamma = 2$ | | | | $\gamma = 3$ | | | | $\gamma = 4$ | | | |
|---|---|---|---|---|---|---|---|---|---|---|---|---|---|---|
| | | | $T_{AR}$ | $T_{SD}$ | $\sigma$ | x | $T_{AR}$ | $T_{SD}$ | $\sigma$ | x | $T_{AR}$ | $T_{SD}$ | $\sigma$ | x |
| 2xH800 | humaneval | 0.0 | 15.96 | 9.34 | 0.95 | 1.71 | 15.96 | 7.95 | 0.93 | 2.01 | 15.96 | 6.96 | 0.90 | **2.29** |
| | humaneval | 1.0 | 17.39 | 12.82 | 0.82 | 1.36 | 13.20 | 8.98 | 0.74 | 1.47 | 13.20 | 7.17 | 0.75 | **1.84** |
| | mtbench | 0.0 | 24.42 | 16.74 | 0.71 | **1.46** | 24.42 | 16.84 | 0.62 | 1.45 | 24.42 | 17.05 | 0.54 | 1.43 |
| | mtbench | 1.0 | 18.24 | 14.38 | 0.67 | **1.27** | 16.25 | 13.28 | 0.56 | 1.22 | 16.25 | 13.76 | 0.48 | 1.18 |
| 4xA800 | humaneval | 0.0 | 11.20 | 6.77 | 0.95 | 1.65 | 11.20 | 5.89 | 0.93 | 1.90 | 11.20 | 5.38 | 0.90 | **2.08** |
| | humaneval | 1.0 | 11.72 | 8.51 | 0.81 | 1.38 | 12.05 | 8.30 | 0.73 | 1.45 | 11.23 | 7.70 | 0.67 | **1.46** |
| | mtbench | 0.0 | 11.26 | 8.92 | 0.72 | **1.26** | 11.26 | 9.10 | 0.61 | 1.24 | 11.26 | 9.82 | 0.52 | 1.15 |
| | mtbench | 1.0 | 11.78 | 10.32 | 0.67 | 1.14 | 11.30 | 9.42 | 0.58 | **1.20** | 11.78 | 11.25 | 0.47 | 1.05 |
| 4xL40 | humaneval | 0.0 | 17.84 | 10.00 | 0.95 | 1.79 | 17.84 | 8.33 | 0.93 | 2.14 | 17.84 | 7.94 | 0.90 | **2.25** |
| | humaneval | 1.0 | 17.89 | 12.27 | 0.80 | 1.46 | 17.89 | 11.07 | 0.74 | 1.62 | 17.89 | 10.91 | 0.65 | **1.64** |
| | mtbench | 0.0 | 20.40 | 15.87 | 0.71 | 1.29 | 20.40 | 16.22 | 0.62 | **1.26** | 20.40 | 16.33 | 0.54 | 1.25 |
| | mtbench | 1.0 | 20.58 | 16.02 | 0.68 | **1.28** | 18.11 | 14.75 | 0.54 | 1.23 | 18.11 | 15.54 | 0.48 | 1.17 |

due to expert loading saturation, and then decreases due to compute-boundness. We denote the maximal speedup across batch sizes as **x** and summarize the results in Table 1. For both models, SD achieves higher acceleration with longer γ for tasks with more predictable patterns (e.g., code generation) or less randomness (e.g., lower temperature), aligning with conclusions from previous research. Figure 5 in Appendix A.1 further presents SD speedup trends under more settings, including individual runs and their mean to show the statistical significance of our findings.

We further evaluate Qwen2-57B-A14B-Instruct on multiple hardware platforms (Table 2) to verify the generality of our conclusions. Combined with results of Qwen2 in Table 1, two observations can be made: (1) GPUs with higher ridge points yield larger SD speedups (e.g., 2×A800 vs. 2×H800, 4×A800 vs. 4×L40), since they provide more arithmetic units for verification. (2) Scaling from 2×A800 to 4×A800 reduces absolute runtimes ($T_{AR}$ and $T_{SD}$), but the SD speedups slightly degrade. This is because the large model benefits from inter-GPU parallelization, whereas the small draft model remains single-GPU, making its relative forward cost higher.

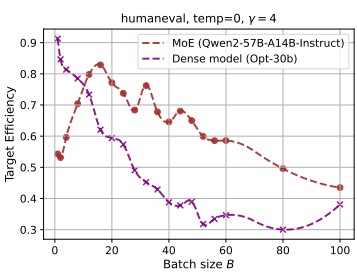

Figure 3: Comparison of target efficiency: MoE vs dense model.

Figure 2 also highlights the effectiveness of our metric target efficiency. It is computed as $\frac{T_T(B,1)}{T_T(B,\gamma)}$ as explained in Sec. 3.1, where both $T_T(B, 1)$ and $T_T(B, \gamma)$ are obtained from vllm runtime logs. Target efficiency values are annotated on the right y-axis, showing a consistent trend with the end-to-end speedup. In contrast, the acceptance rate across batch sizes merely *fluctuates within a small range*, unable to effectively explain the dramatic changes in speedup.

We further compare MoE and dense models via target efficiency (whose validity is established above) to avoid interference from acceptance rate. As shown in Figure 3, the target efficiency for MoE first increases and then decreases, while that for the dense model decreases continuously. Consequently, although SD for MoE is less effective with small batches, it exhibits stronger potential across a wider range of larger batch sizes.

Regarding end-to-end performance, SD speedups become more pronounced for MoE when the batch size exceeds 16, as supplemented in Figure 6 in Appendix A.2.

## 4.2 Impact of MoE Sparsity and Validation of Modeling Method

To evaluate MoE sparsity's impact on SD acceleration, we varied the number of activated experts per token ($K$) of Qwen2-57B-A14B-Instruct. Directly changing $K$ without training affects the target model's performance and speculation accuracy, so we adjust the speedup by multiplying the raw speedup with $\frac{\sigma_{K=8}}{\sigma_K}$, whose rationale is exhibited by Eq. 4. Fig. 4 shows the adjusted speedup alongside our modeling results for comparison. The parameters used in the modeling are decided using 21 GPU measurements, as explained in Section 3.3. The impact of measurement selection for parameter fitting on the modeling's reliability is supplemented in Appendix C.

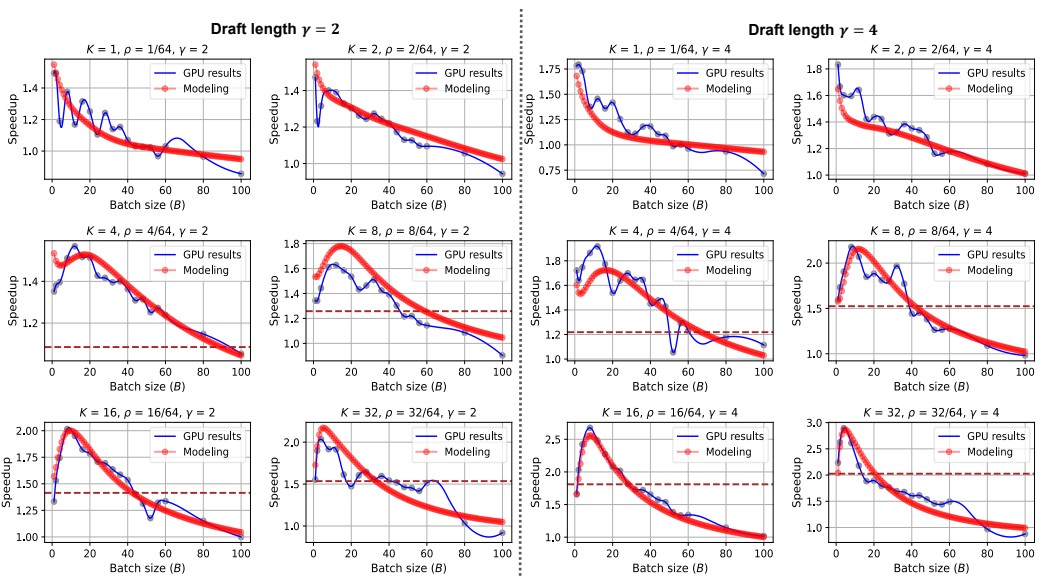

Figure 4: Comparison between GPU results and our modeling for Qwen2-57B-A14B-Instruct with varying sparsity $\rho$ and draft length $\gamma$.

There are three key observations. First, the modeling consistently aligns with experiment results across varying sparsity ($\rho$) and draft length ($\gamma$), validating our modeling's reliability.

Second, while the SD speedup in most MoEs exhibits an initial increase followed by a decrease, very sparse MoEs ($K = 1, 2$) show continuously decreasing speedup. This appears to conflict with the theoretical analysis, but after examining the components of our modeling, we identified the reason as follows. These very sparse MoEs have a disproportionately low ratio of FFN, thus making the memory-boundness of MoE FFN hard to manifest systematically as indicated by Amdahl's Law. The Qwen2-57B-A14B model is designed based on $K = 8$, but by reducing $K$ to 1 or 2, we actually artificially *synthesize* a model where Attention dominates. In practice, however, sparser MoEs typically incorporate more FFN parameters to maintain a balanced ratio between FFN and Attention components, resulting in acceleration patterns more similar to $K = 8$ cases.

Finally, as MoE models become sparser, the system's transition from memory-bound to compute-bound is delayed. This is evidenced by two phenomena in Fig. 4: With smaller $\rho$, (1) the batch size for the maximal speedup ($\mathbf{x}$) becomes larger; (2) the range of batch sizes that maintain speedup above a certain decay threshold (annotated by the brown dashed line in Fig. 4 for $\mathbf{x}/\sqrt{2}$) is wider. These validate our theoretical analysis and indicate that SD has broader applicability in sparser MoEs.

## 5   Conclusion and Limitation

This work challenges the conventional wisdom that speculative decoding (SD) cannot effectively accelerate MoE models. We demonstrate that under moderate batch sizes, sparser MoEs gain greater speedup from SD due to their memory-bound FFNs, a conclusion supported by both theoretical and experimental analysis. To navigate the complex factors influencing speedup, we develop a reliable, interpretable SD model and introduce target efficiency to elucidate the impact of model architecture and workload. Our work offers a new perspective for accelerating MoEs, particularly in memory-constrained or moderate-batch serving. While our analysis assumes the KV-cache is smaller than parameters, the behavior when it dominates is analyzed by MagicDec [34]; these two works combine to give a more comprehensive view of SD across batch sizes.

## Acknowledgments and Disclosure of Funding

This work is supported by the National Key Research and Development Program of China under Grant (2024YFB4405400).

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

Figure 5: SD speedup trends across more settings with individual runs and averages shown.

## A  Supplementary Experimental Results

This section presents additional experimental results referenced in Section 4, which are included here due to space limitations.

### A.1  Trends of SD speedup under more configurations

Figure 5 presents additional trends of SD speedup across different datasets, temperatures, and model types, serving as a supplement to Figure 2. The results demonstrate that SD speedup exhibits a consistent first-increase-then-decrease pattern, which aligns well with our theoretical analysis.

To confirm the statistical significance of our findings, we also present the five individual runs that constitute the averages in Figure 5. The variance across different runs is minimal, which is expected since the random seed is fixed across all runs to ensure identical workloads.

While the overall trend follows the first-increase-then-decrease pattern, local fluctuations are observable in the curves. For instance, Figure 5(c) exhibits a sawtooth-like decreasing trend. This phenomenon can be attributed to GPU *quantization effects*, as documented in NVIDIA's documentation [47]. When dimensions are not evenly divisible by the GPU's native tile sizes, computational performance degrades. AR decoding is more sensitive to this effect than SD, making the time ratio of

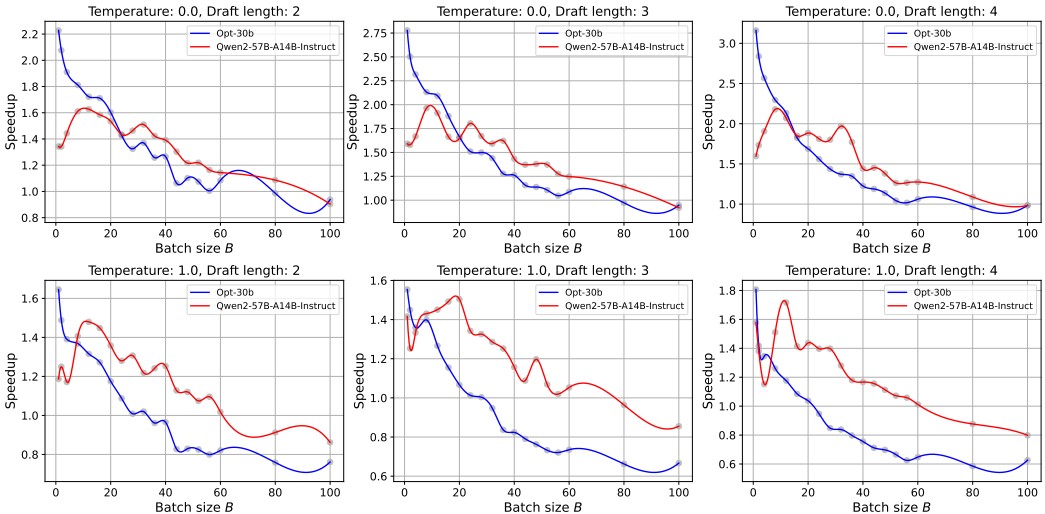

Figure 6: End-to-end speedup comparison of MoE and dense models under various settings.

AR to SD (namely, SD speedup) fluctuate. Despite these local variations, the overall speedup trend follows our theoretical predictions, confirming the validity of our conclusions.

## A.2  End-to-end speedup comparison between MoE and dense models

To isolate the effects of acceptance rate variations and enable a clearer focus on system bottlenecks, we have compared MoE and dense models using target efficiency in Section 4.1. In this section, we further compare their end-to-end speedup across various settings in Figure 6 as a supplement.

Two key observations emerge from the experiment results. First, while SD speedups for MoE models initially increase before declining, SD speedups for dense models continue to decrease. Consequently, SD achieves more substantial end-to-end speedups for MoE models at moderate batch sizes, which aligns with the trend in Figure 3 in Section 4.1. Second, the extent to which SD favors MoE over dense models varies across different configurations. For instance, at temperature = 1 (second row), SD demonstrates greater relative benefits for MoE compared to temperature = 0 (first row). This variation stems from diverse acceptance rates under different settings, which can obscure the observation of systemic bottlenecks. In summary, target efficiency serves as a reliable comparison metric while controlling for the confounding effects of algorithmic optimizations.

# B  Proof of $\overline{T_{exp}}(T; \rho)$'s Trend with Varying $\rho$

In Section 3.2, Fig. 1(c) demonstrates that: Given input token count $t = T > 1$, the number of tokens each expert processes on average $\overline{T_{\exp}}(T; \rho) = \frac{\rho T}{1-(1-\rho)^T}$ decreases as $\rho$ decreases. We prove this by showing $\frac{d(\overline{T_{\exp}}(T;\rho))}{d\rho} > 0$ when $T > 1$.

$$\frac{d(\overline{T_{\exp}}(T; \rho))}{d\rho} = \frac{d(\frac{\rho T}{1-(1-\rho)^T})}{d\rho} = \frac{T(-\rho T(1-\rho)^{T-1} - (1-\rho)^T + 1)}{(1-(1-\rho)^T)^2} \tag{12}$$

Since $\rho$ represents MoE sparsity $\in (0, 1)$, the original proposition is equivalent to proving:

$$\mathbf{F}(\rho; T) = (1-\rho)^{T-1}(\rho T + 1 - \rho) < 1 \tag{13}$$

Note that $\mathbf{F}(\rho; T) \to 1$ as $\rho \to 0$. Therefore, if we can prove that $\mathbf{F}(\rho; T)$ decreases as $\rho$ increases from 0 to 1, then the original proposition is proven. We demonstrate this by computing $\frac{d(\mathbf{F}(\rho; T))}{d\rho}$:

$$\frac{d(\mathbf{F}(\rho; T))}{d\rho} = \frac{d((1-\rho)^{T-1}(\rho T + 1 - \rho))}{d\rho} = -\rho T(T-1)(1-\rho)^{T-2} \tag{14}$$

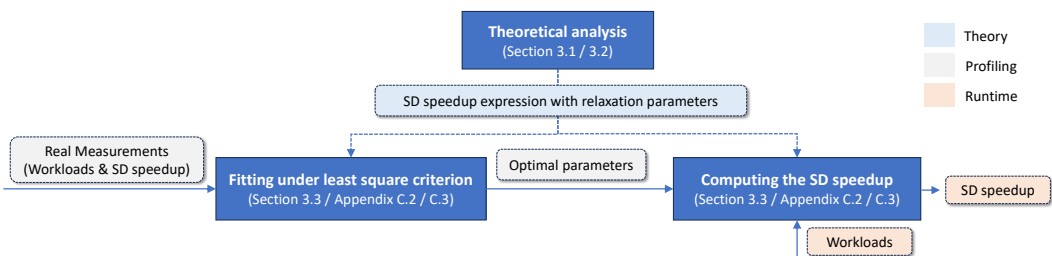

Figure 7: The overall diagram of the modeling method.

When $T > 1$, $\frac{\mathrm{d}(\mathbf{F}(\rho;T))}{\mathrm{d}\rho} < 0$. This confirms that $\mathbf{F}(\rho;T)$ decreases as $\rho$ increases, which proves our original proposition: when $T > 1$, $\overline{T_{\exp}}(T;\rho)$ decreases as $\rho$ decreases.

# C More details about the Modeling Method

The main design considerations and expressions of our modeling method have been presented in Section 3.3. In this section, we provide additional content on the following topics to give a more comprehensive view of the modeling method:

- Description and an illustrative diagram of the modeling process. (Appendix C.1)
- Fitting Details of the modeling shown in Figure 4 in Section 4.1. (Appendix C.2)
- How the modeling is affected by alternative measurement selection. (Appendix C.3)

The value of our modeling is twofold. On one hand, it achieves alignment with real measurements with only a small number of simple parameters, thus validating the correctness of our theoretical analyses. On the other hand, it provides the decomposition of various factors in the end-to-end results, making the entire SD acceleration process explainable and transparent.

## C.1 Description and Overview of the Modeling Process

Figure 7 presents the overall diagram of our modeling method. Building upon the theoretical analyses in Sections 3.1 and 3.2, we derive an expression for SD speedup as a function of workloads. This expression contains several relaxation parameters to be determined for complete modeling. We determine these parameters through empirical profiling. We first collect a small set of real measurements comprising various workloads and their corresponding SD speedups. We then perform parameter fitting using these measurements under the least squares criterion to obtain optimal parameter values. Since our theoretically-derived SD speedup expression already captures the fundamental performance tradeoffs, the fitting process is computationally lightweight and robust, as will be demonstrated in Appendix C.2 and C.3. Once the optimal parameters are obtained, the resulting expression can predict SD speedups for arbitrary workloads.

We now explain why profiling is necessary and why we cannot derive the complete SD speedup expression purely through theoretical analysis, examining both software and hardware considerations.

**Software considerations:** Actual execution times can deviate significantly from theoretical predictions for complex operators with diverse implementations. On GPUs, GEMM operations are indeed predictable due to their regular structure and highly optimized implementations. However, prediction becomes challenging for operators such as Attention, which involve customized kernel optimizations (e.g., FlashAttention1/2, eager attention, SDPA attention) and operator fusion strategies (incorporating various nonlinear layers or positional encodings such as RoPE and its variants). To illustrate this complexity, we examine profiling results from Qwen2-57B-A14B (hidden size 3584) and Mixtral-8x7B (hidden size 4096). For FFN, Qwen takes a shorter time than Mixtral (143us vs 226us), aligning with their relative hidden sizes. However, for Attention, Qwen takes a longer time than Mixtral (271us vs 115us), contradicting the theoretical expectation based on hidden size scaling.

**Hardware considerations:** GPU microarchitectures vary across different series, which can greatly impact execution times. For instance, attention efficiency depends heavily on hardware-aware

programming optimizations, while different GPUs vary in cache configurations and thread-memory interaction patterns across memory hierarchy levels. By taking advantage of new capabilities in modern hardware, FlashAttention-3 successfully increases GPU utilization from 35% to 75% on H100 GPUs [48]. Moreover, many critical hardware details remain undisclosed by GPU vendors, making theoretical predictions impractical.

Therefore, modeling speedup trends with pure analytical methods requires *case-by-case* analysis for different operator implementations and GPU microarchitectures. In contrast, the hyperparameter approach offers a more generalizable paradigm and is easy to use: all parameters possess clear physical interpretations, only minimal profiling data are required, and the computational overhead is low. Our method achieves a balance between effectiveness and practicality: on one hand, it captures the primary performance drivers (i.e. the number of activated experts and roofline trends); on the other hand, it avoids getting entangled in low-level implementation complexities. This approach is also used by other system optimization frameworks such as NanoFlow [49], which similarly adopt a two-stage strategy of profiling followed by runtime execution.

### C.2 Fitting Details of the Modeling shown in Figure 4

We first explain how we select the 21 measurements. Due to GPU resource and time constraints, we obtained a total of 228 GPU measurements across varying experimental settings, including 6 different numbers of activated experts per token ($K$), 2 draft lengths ($\gamma$), and 19 batch sizes ($B$). These measurements are sorted first by $K$, then by $\gamma$ within each $K$ group, and finally by $B$ within each $(K, \gamma)$ combination, forming the total dataframe (`df`). We then uniformly sampled measurements from this sorted dataset with a fixed stride, namely `M = df[begin:end:11]`. This sampling strategy enables our selected measurements to contain different settings, making the modeling more robust.

The SD speedup function (namely, *ComputeSpeedup* defined in line 3 of Algorithm 1) is nonlinear. To optimize its MSE, we employed the `scipy.optimize.least_squares` function with the Trust Region Reflective (TRR) algorithm. TRR is an optimization method for bound-constrained nonlinear least squares problems that combines trust region methods with reflection techniques. It constructs quadratic models within trust regions and uses reflection strategies near boundaries to maintain feasibility while ensuring convergence. The fitting process for these 21 data points is efficient, completing in approximately 0.114 seconds. Our modeling incorporates 10 parameters requiring relaxation, with their search boundaries specified as follows:

- *bias*: It represents the time required to load the dense parameters of the target model. We denote the model's non-FFN parameter count as $V_{dense}$. Consequently, the theoretical minimum loading time can be calculated as $bias_{min} = \frac{V_{dense} \times bitwidth}{peak\ memory\ bandwidth}$. For the upper bound of the relaxation range, we set $bias_{max} = 5 \times bias_{min}$.

- *k1*: It adjusts the intensity of the roofline effect of dense components. It should be larger than 0 to ensure the execution time increases as the token count increases. We don't set a definite upper limit for $k1$, as its value is affected by other parameters. Given the hardware with fixed arithmetic units, the execution time grows linearly with the token count in the compute-bound regime. As shown in line 6 of Algorithm 1, $k1$ appears as a coefficient in the term $k1 \cdot G(t; \lambda, s)$, whose gradient in the compute-bound regime is $k1 \cdot ln(s) \cdot s^{\lambda RP}$. As $s$ approaches 1, $k1$ needs to continuously increase to counterbalance $ln(s)$ that approaches 0.

- *k2*: It represents the time required to load one expert. Given a target model, we denote the parameter count per expert as $V_{exp}$. Consequently, the theoretical minimum loading time can be calculated as $k2_{min} = \frac{V_{exp} \times bitwidth}{peak\ memory\ bandwidth}$. For the upper bound of the relaxation range, we set $k2_{max} = 5 \times k2_{min}$.

- *k3*: It adjusts the intensity of the roofline effect of sparse components. Similar to *k1*, we set $k3_{min} = 0$ and $k3_{max} = inf$.

- *draft_bias*: It represents the time required to load the dense draft model. We denote the draft model's parameter count as $V_{draft}$. Consequently, the theoretical minimum loading time can be calculated as $draft\_bias_{min} = \frac{V_{draft} \times bitwidth}{peak\ memory\ bandwidth}$. For the upper bound of the relaxation range, we set $draft\_bias_{max} = 5 \times draft\_bias_{min}$.

- *draft_k*: It adjusts the intensity of the roofline effect of the dense draft model. Similar to *k1*, we set $draft\_k_{min} = 0$ and $draft\_k_{max} = inf$.

- *reject_bias*: It represents the fixed overhead when performing rejection sampling. Vllm reports its elapsed time during SD, and we denote the maximum across measurements as $T_{rej}$. We then set *reject_bias*$_{min} = 0$ and *reject_bias*$_{max} = T_{rej}$.

- *reject_k*: It represents the incremental processing time in rejection sampling as the input token count increases. For simplicity, we set *reject_k*$_{min} = 0$ and *reject_k*$_{max} = T_{rej}$ just like *reject_bias*.

- $\lambda$: It represents the ratio of the empirical ridge point to the theoretical ridge point. Since memory bandwidth is typically less utilized than arithmetic units, we set $\lambda_{min} = 0.2$ and $\lambda_{max} = 1$.

- $s$: It adjusts the growing rate of execution time as input token count increases. Since $s$ serves as the base of $G(t)$, it must exceed 1 to ensure monotonic growth. However, $s$ should not be too large, as it would result in an excessively steep growth rate. In experiments, we set $s_{min} = 1$ and $s_{max} = 2$.

## C.3 Exploration of Alternative Measurement Selection

In this section, we demonstrate the impact of varying the number ($m$) of measurements used for fitting on the modeling results. Given that our model incorporates 10 parameters, a minimum of 10 profiling data points ($m \geq 10$) are required to determine all parameters. We present the modeling fitting with $m$ ranging from 10 to 228. The data selection method follows the stride-based approach described in the previous section, specifically M = df[begin:end:$stride$]. Measurement count $m$ and $stride$ satisfy the following relation: $m = \lceil 228/stride \rceil$.

We present the MSE values of different $m$s and their corresponding fitting figures in Table 3. We also list the distinct batch sizes involved in the selected measurements, which helps explain why some configurations show inferior model fit. Due to integer division constraints, $m$s are not continuous at larger magnitudes. Generally, the modeling fits well with the real measurements, except for $m = 10, 12, 13$. The reasons are as follows. When $m = 10$, the number of measurements equals the parameter count, resulting in insufficient data for robust fitting. When $m = 12$ and $m = 13$, the distribution of the measurement data is biased. With stride-based selection, measurements at $m = 12$ and $m = 13$ demonstrate notable gaps in batch size coverage (specifically, $m = 12$ does not include batch sizes greater than 40, while $m = 13$ does not include batch sizes within 1~24). Their MSE values are larger than that of $m = 11$, despite the latter containing fewer data points for fitting. Based on this analysis, we recommend prioritizing uniform data distribution when selecting measurements, as this approach enables the development of more reliable models even with smaller datasets.

Table 3

| $m$ | $stride$ | MSE | Figure | Batch Size Involved |
|---|---|---|---|---|
| 10 | 25 | 2.216 | 8 | [1, 12, 16, 20, 36, 40, 44, 60, 80, 100] |
| 11 | 22 | 1.764 | 9 | [1, 4, 8, 16, 20, 28, 32, 40, 44, 56, 100] |
| 12 | 20 | 4.288 | 10 | [1, 2, 4, 8, 12, 16, 20, 24, 28, 32, 36, 40] |
| 13 | 18 | 2.681 | 11 | [1, 24, 28, 32, 36, 40, 44, 48, 52, 56, 60, 80, 100] |
| 14 | 17 | 2.041 | 12 | [1, 2, 8, 16, 24, 32, 40, 44, 48, 52, 56, 60, 80, 100] |
| 15 | 16 | 1.668 | 13 | [1, 2, 4, 12, 16, 24, 28, 36, 40, 48, 52, 56, 60, 80, 100] |
| 16 | 15 | 1.508 | 14 | [1, 2, 4, 8, 16, 20, 24, 32, 36, 40, 48, 52, 56, 60, 80, 100] |
| 17 | 14 | 1.563 | 15 | [1, 2, 4, 8, 12, 20, 24, 28, 32, 40, 44, 48, 52, 56, 60, 80, 100] |
| 18 | 13 | 1.525 | 16 | [1, 2, 4, 8, 12, 16, 24, 28, 32, 36, 40, 44, 48, 52, 56, 60, 80, 100] |
| 19 | 12 | 2.080 | 17 | [1, 2, 4, 8, 12, 16, 20, 24, 28, 32, 36, 40, 44, 48, 52, 56, 60, 80, 100] |
| 21 | 11 | 1.679 | 18 | [1, 2, 4, 8, 12, 16, 20, 24, 28, 32, 36, 40, 44, 48, 52, 56, 60, 80, 100] |
| 23 | 10 | 1.800 | 19 | [1, 2, 4, 8, 12, 16, 20, 24, 28, 32, 36, 40, 44, 48, 52, 56, 60, 80, 100] |
| 26 | 9 | 1.716 | 20 | [1, 2, 4, 8, 12, 16, 20, 24, 28, 32, 36, 40, 44, 48, 52, 56, 60, 80, 100] |
| 29 | 8 | 1.524 | 21 | [1, 2, 4, 8, 12, 16, 20, 24, 28, 32, 36, 40, 44, 48, 52, 56, 60, 80, 100] |
| 33 | 7 | 1.526 | 22 | [1, 2, 4, 8, 12, 16, 20, 24, 28, 32, 36, 40, 44, 48, 52, 56, 60, 80, 100] |
| 38 | 6 | 1.715 | 23 | [1, 2, 4, 8, 12, 16, 20, 24, 28, 32, 36, 40, 44, 48, 52, 56, 60, 80, 100] |
| 46 | 5 | 1.644 | 24 | [1, 2, 4, 8, 12, 16, 20, 24, 28, 32, 36, 40, 44, 48, 52, 56, 60, 80, 100] |
| 57 | 4 | 1.509 | 25 | [1, 2, 4, 8, 12, 16, 20, 24, 28, 32, 36, 40, 44, 48, 52, 56, 60, 80, 100] |
| 76 | 3 | 1.553 | 26 | [1, 2, 4, 8, 12, 16, 20, 24, 28, 32, 36, 40, 44, 48, 52, 56, 60, 80, 100] |
| 114 | 2 | 1.485 | 27 | [1, 2, 4, 8, 12, 16, 20, 24, 28, 32, 36, 40, 44, 48, 52, 56, 60, 80, 100] |
| 228 | 1 | 1.523 | 28 | [1, 2, 4, 8, 12, 16, 20, 24, 28, 32, 36, 40, 44, 48, 52, 56, 60, 80, 100] |

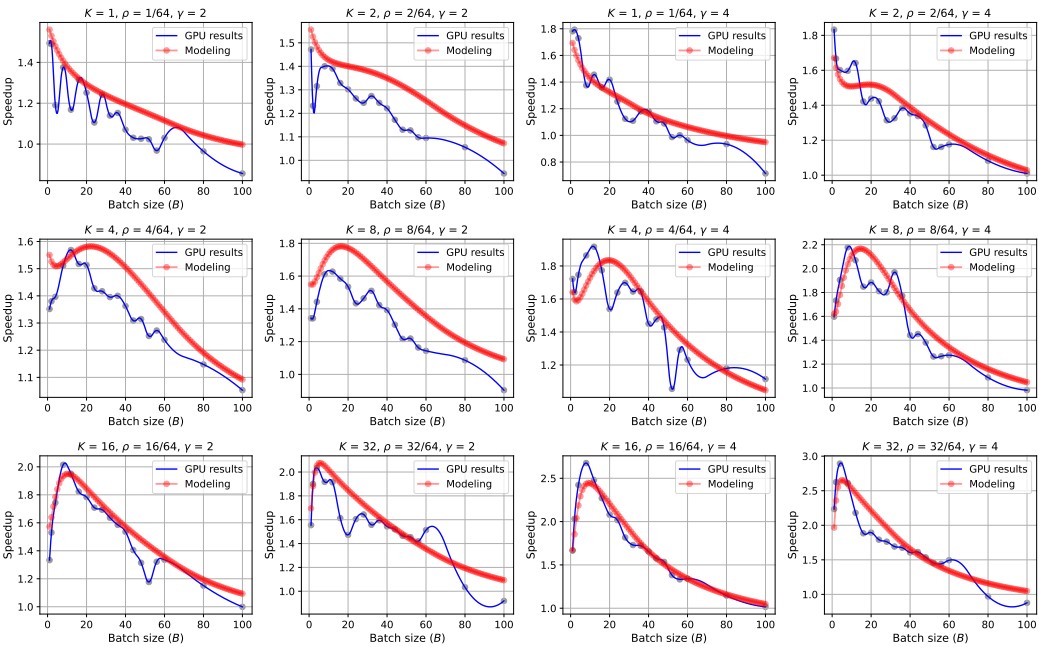

Figure 8: Comparison between GPU results and modeling with 10 measurements.

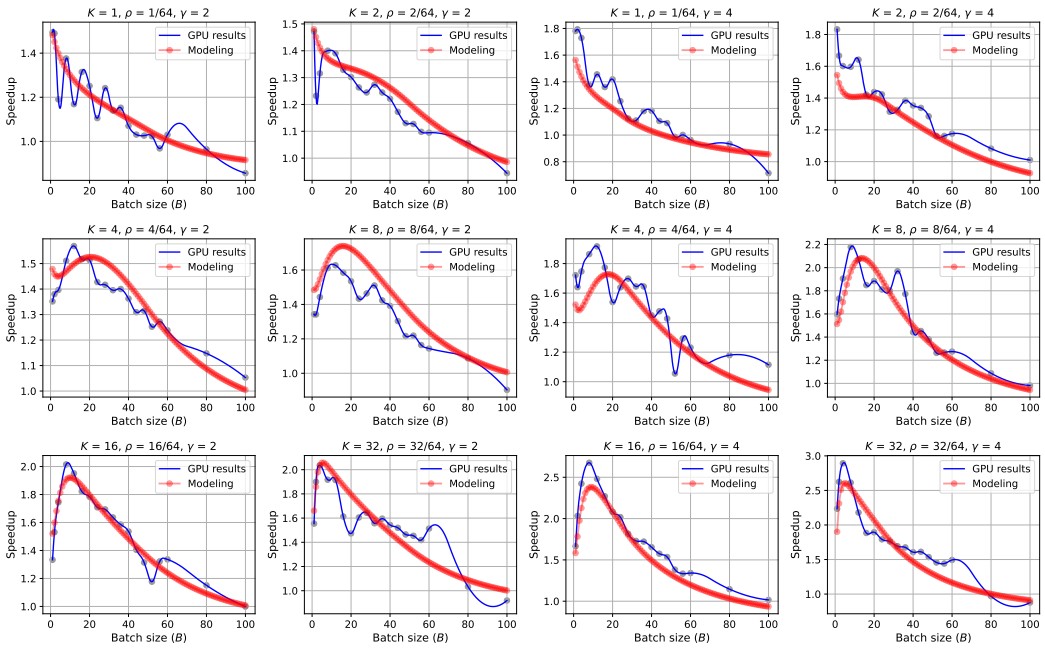

Figure 9: Comparison between GPU results and modeling with 11 measurements.

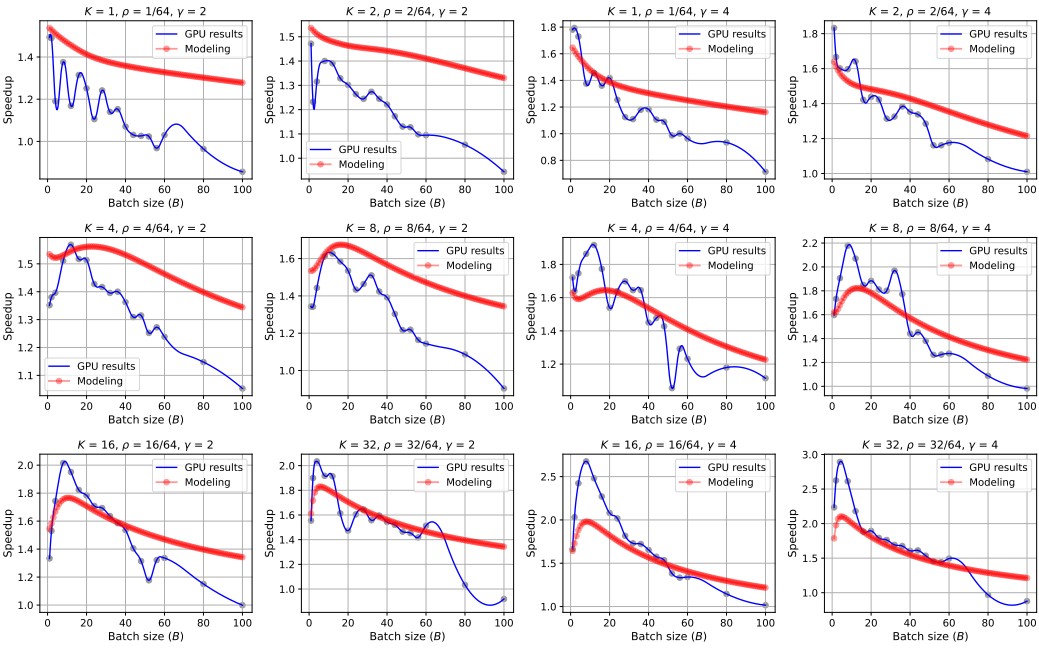

Figure 10: Comparison between GPU results and modeling with 12 measurements.

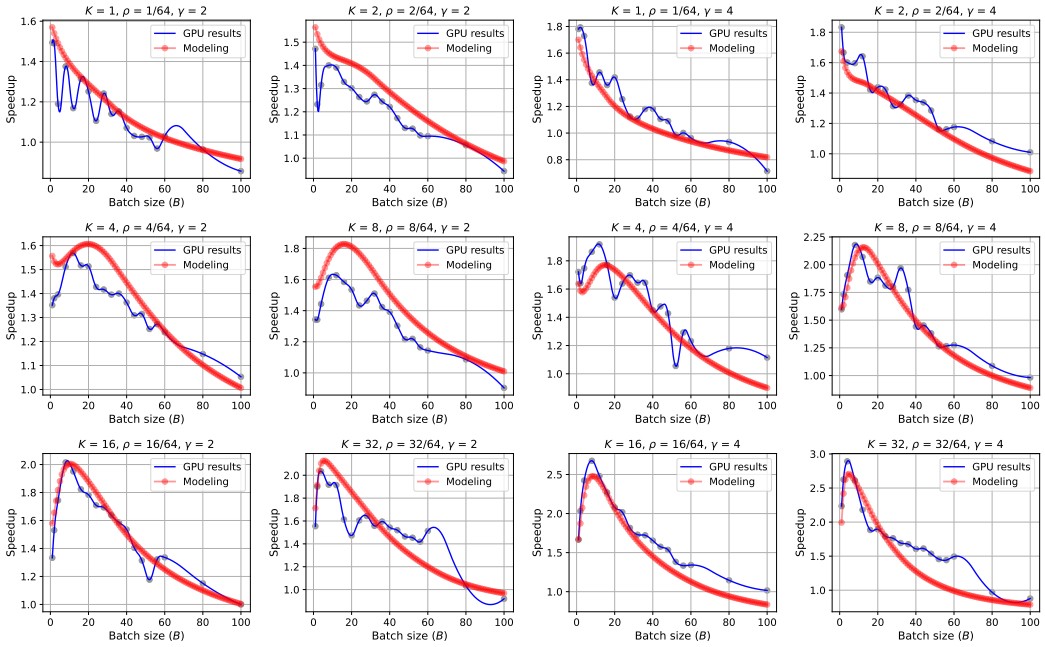

Figure 11: Comparison between GPU results and modeling with 13 measurements.

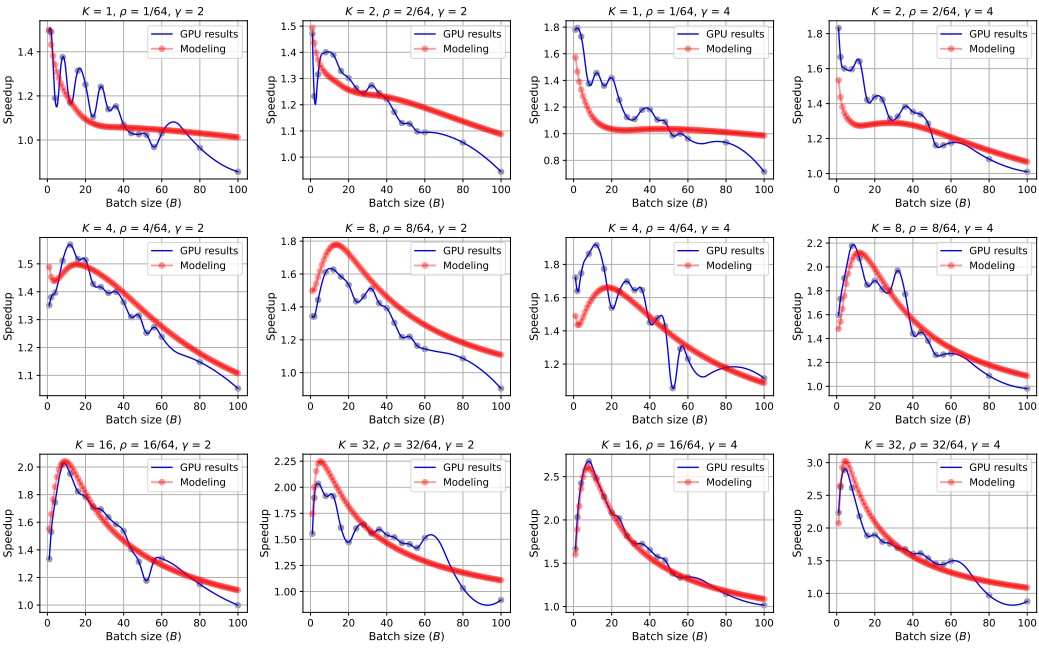

Figure 12: Comparison between GPU results and modeling with 14 measurements.

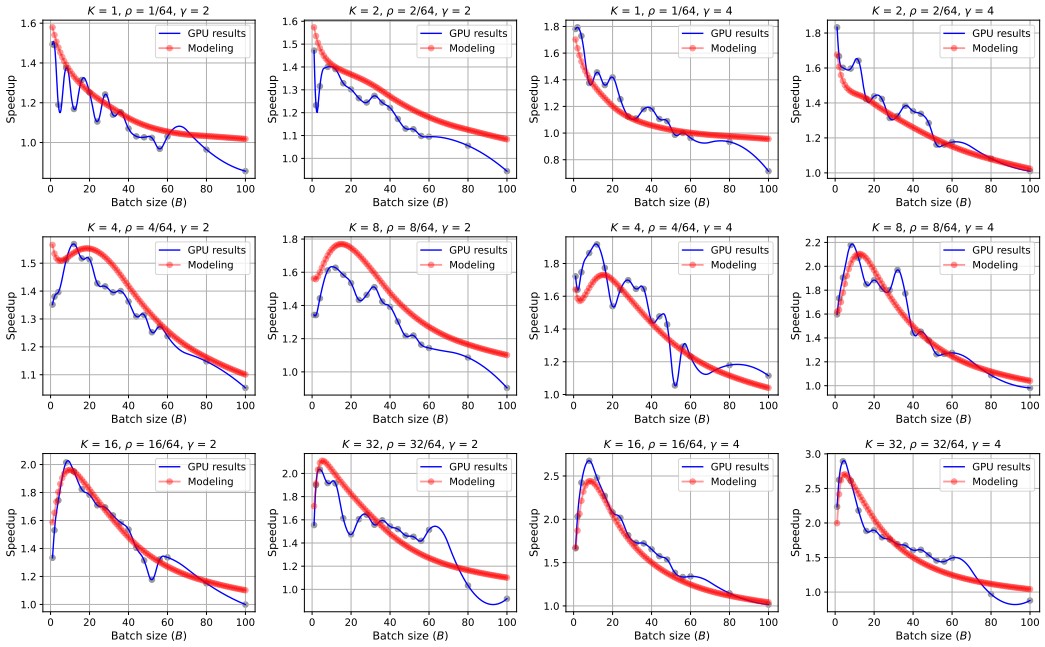

Figure 13: Comparison between GPU results and modeling with 15 measurements.

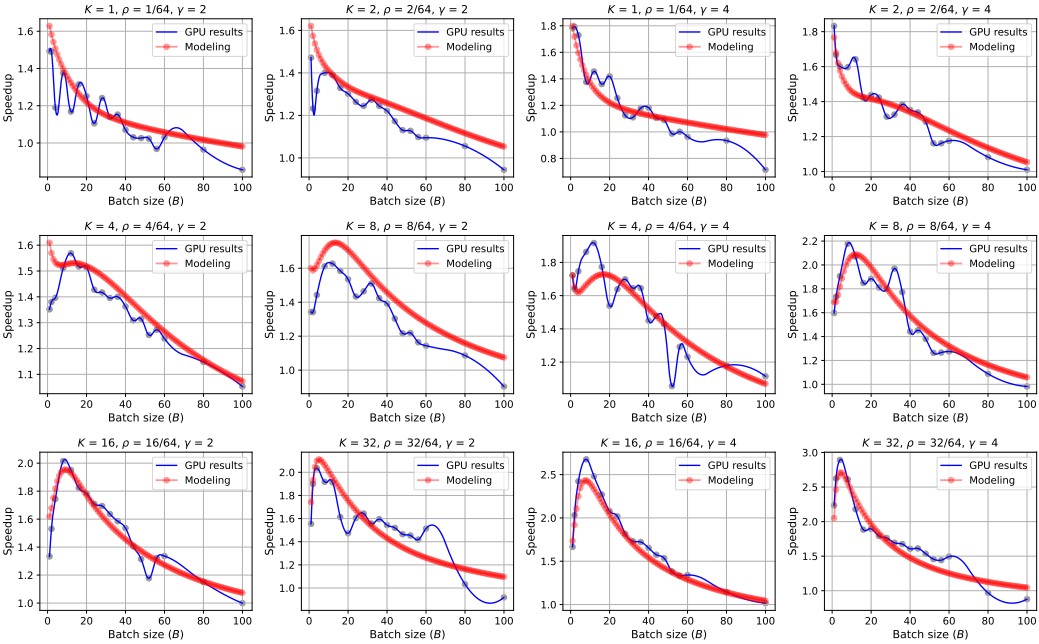

Figure 14: Comparison between GPU results and modeling with 16 measurements.

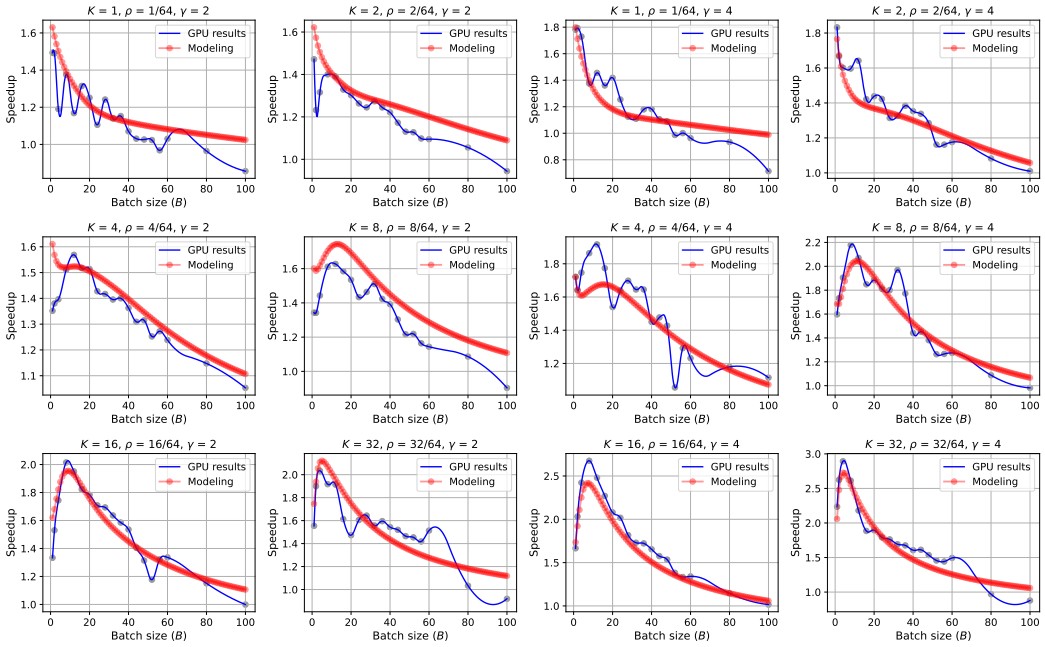

Figure 15: Comparison between GPU results and modeling with 17 measurements.

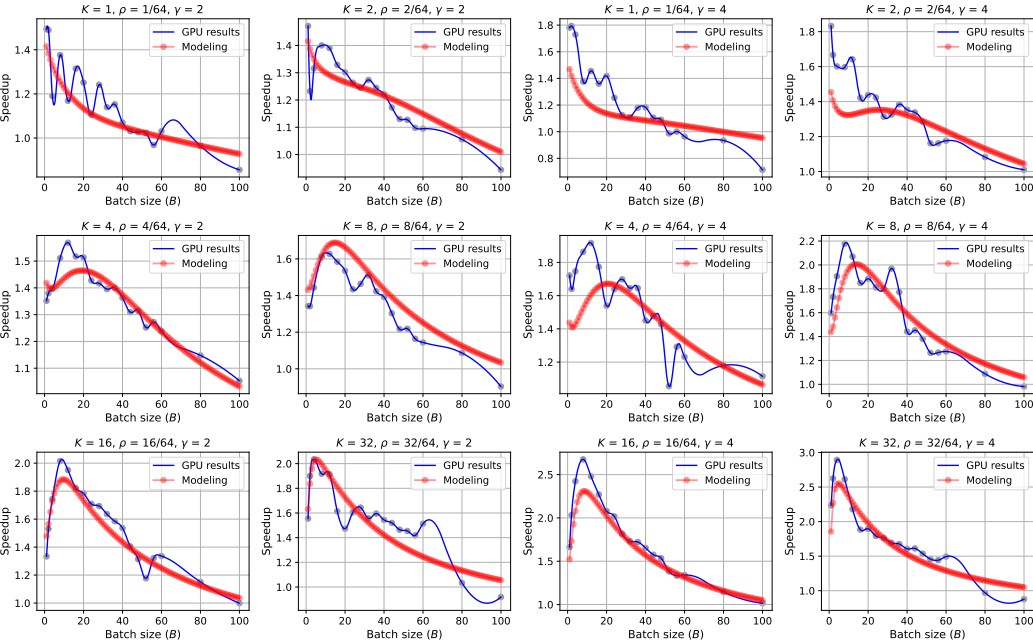

Figure 16: Comparison between GPU results and modeling with 18 measurements.

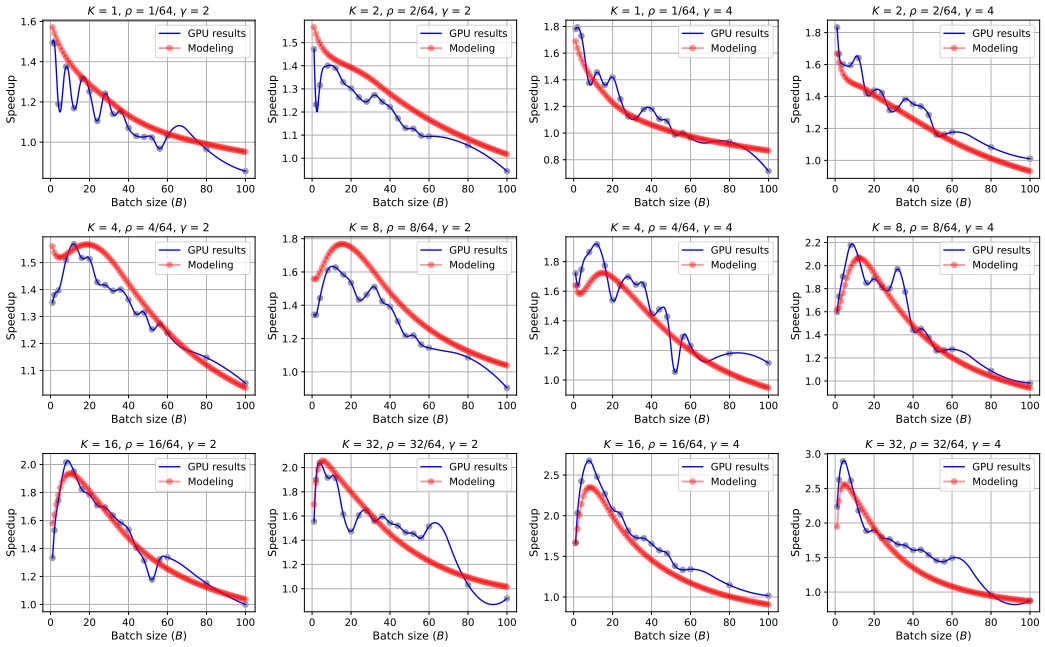

Figure 17: Comparison between GPU results and modeling with 19 measurements.

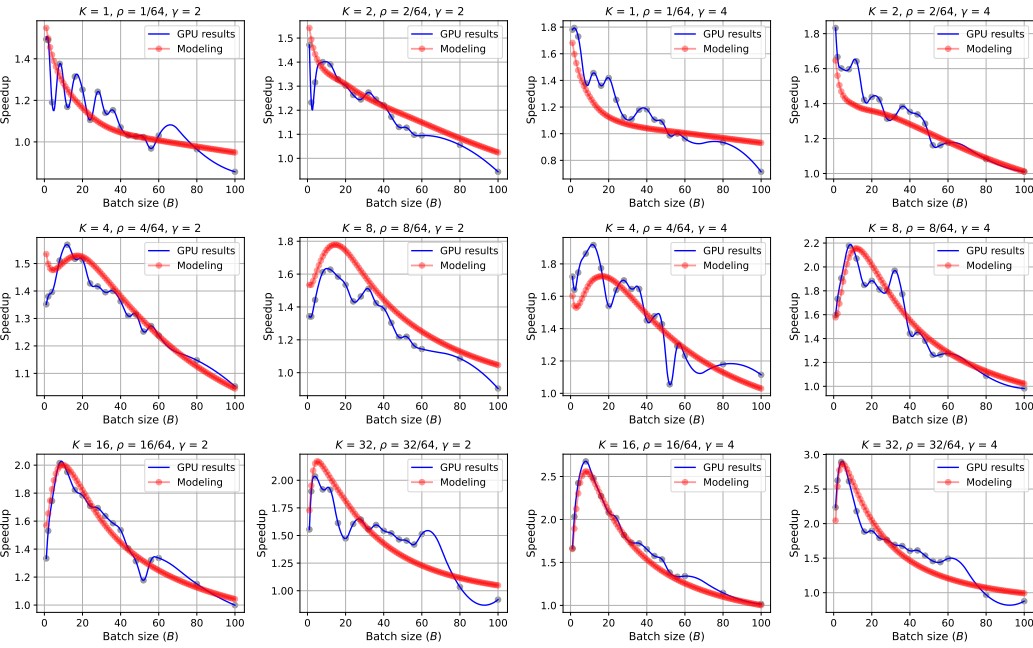

Figure 18: Comparison between GPU results and modeling with 21 measurements.

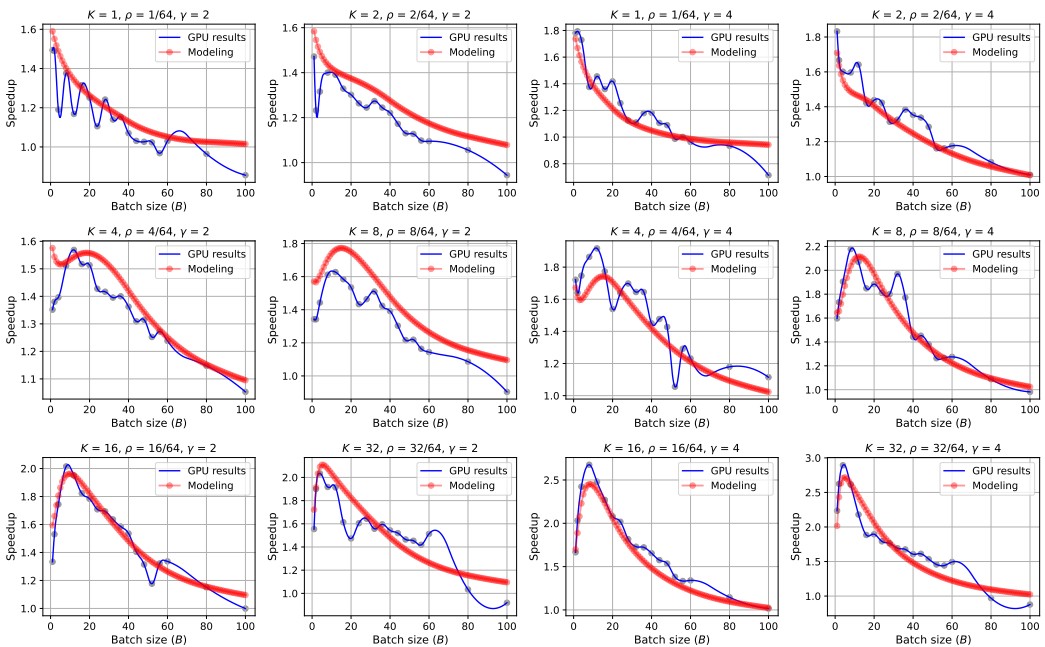

Figure 19: Comparison between GPU results and modeling with 23 measurements.

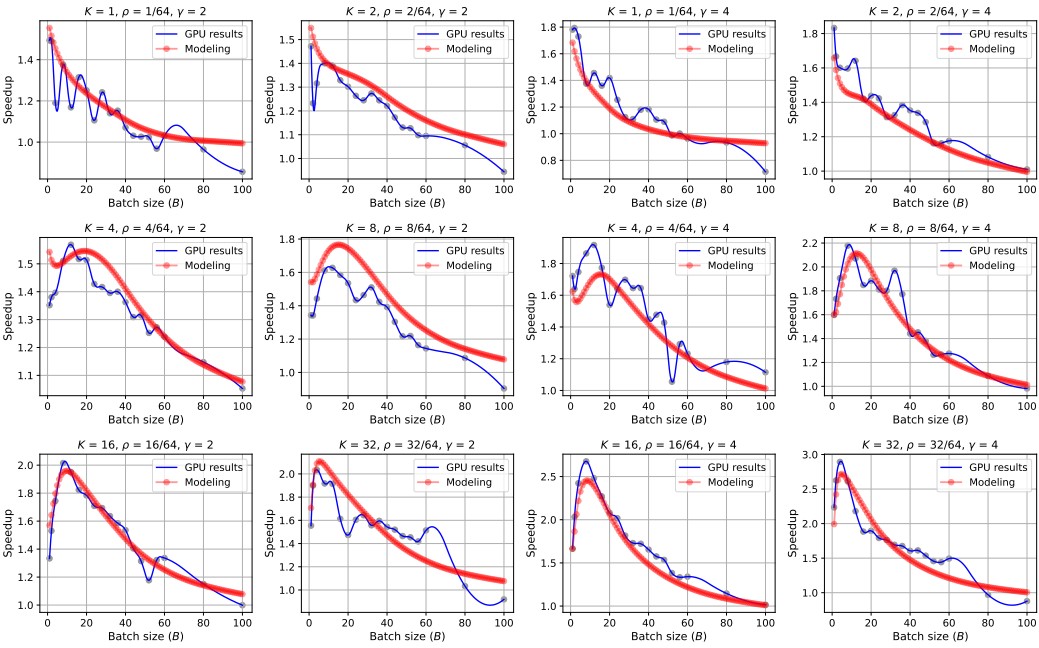

Figure 20: Comparison between GPU results and modeling with 26 measurements.

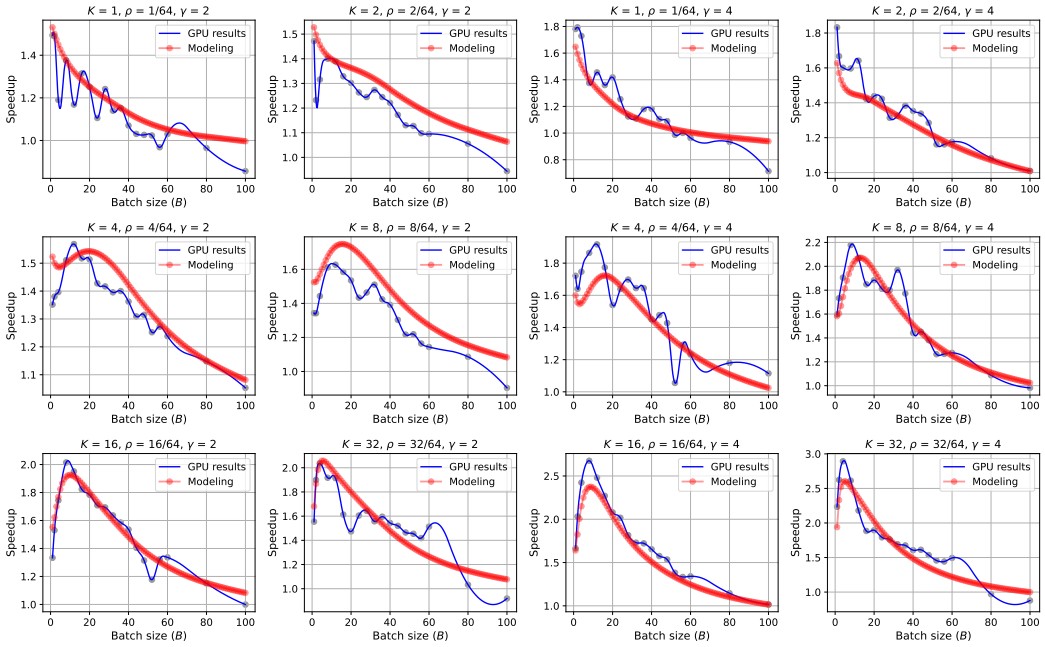

Figure 21: Comparison between GPU results and modeling with 29 measurements.

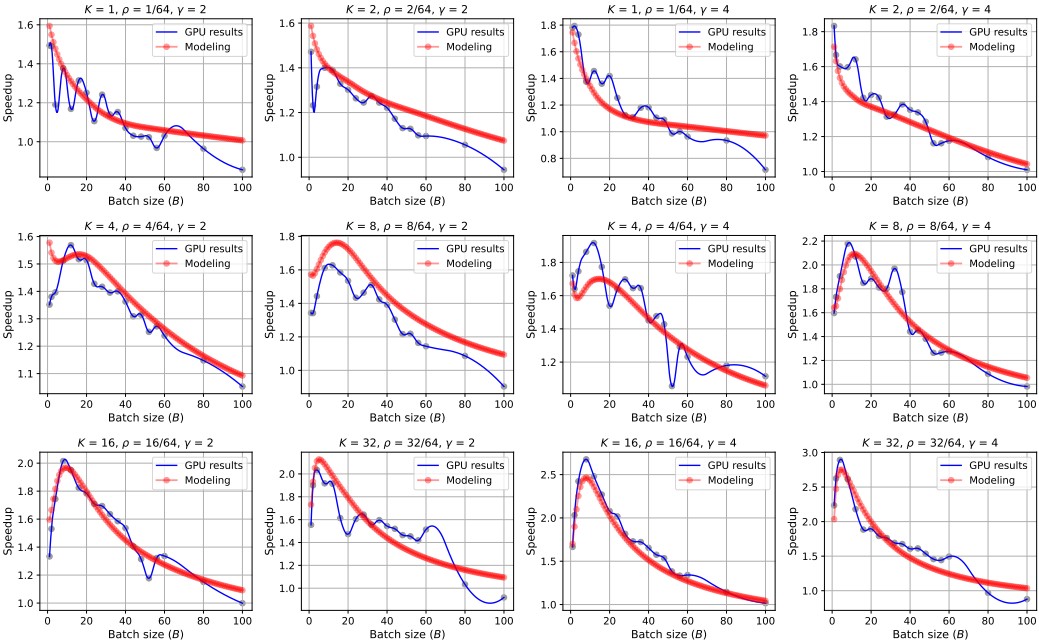

Figure 22: Comparison between GPU results and modeling with 33 measurements.

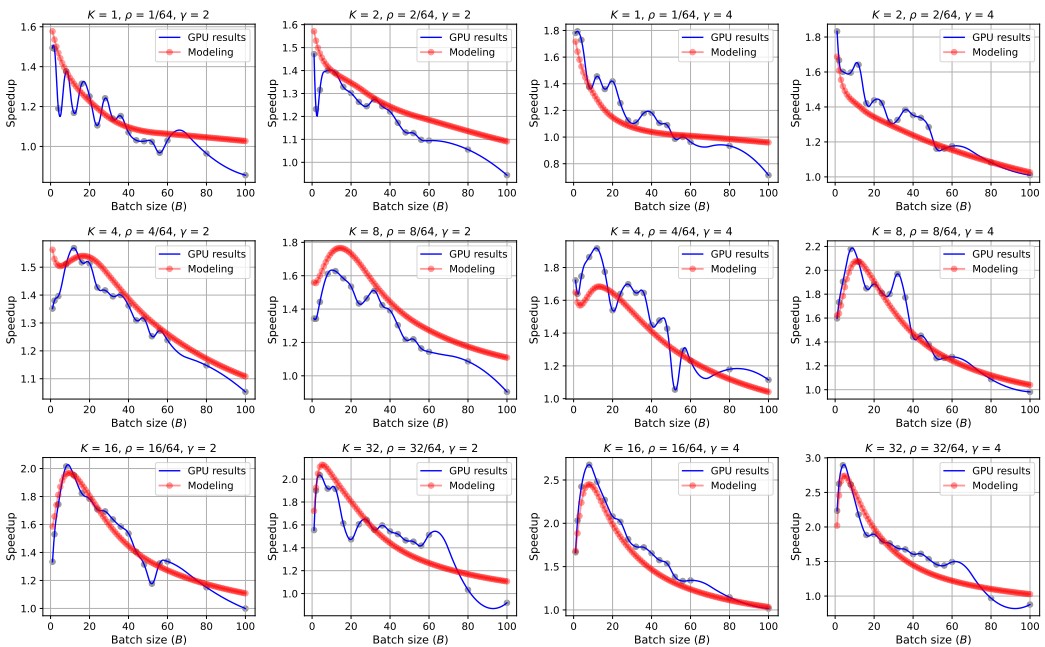

Figure 23: Comparison between GPU results and modeling with 38 measurements.

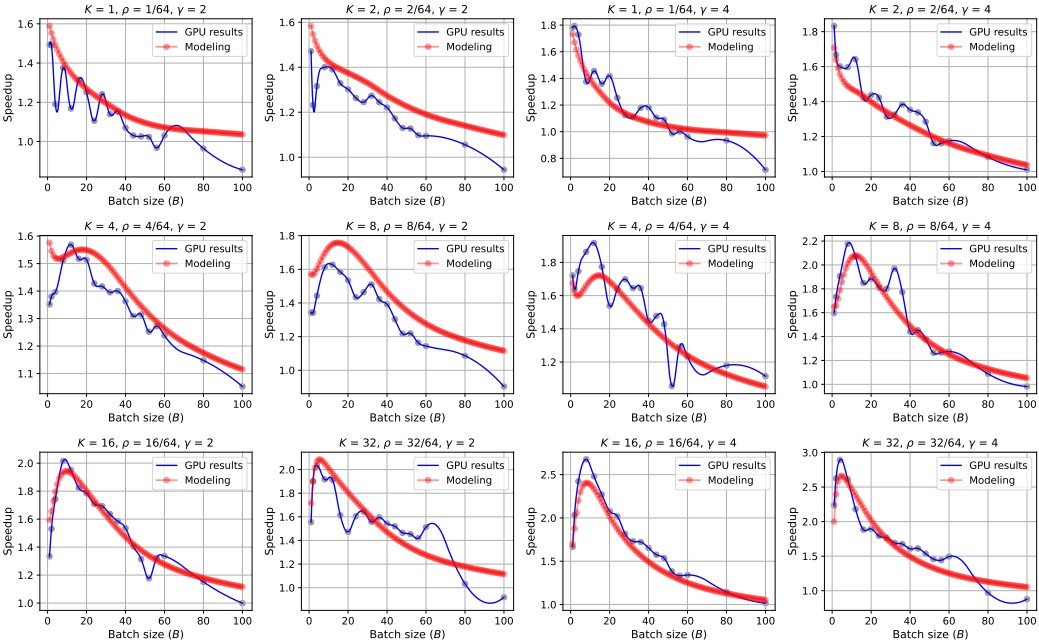

Figure 24: Comparison between GPU results and modeling with 46 measurements.

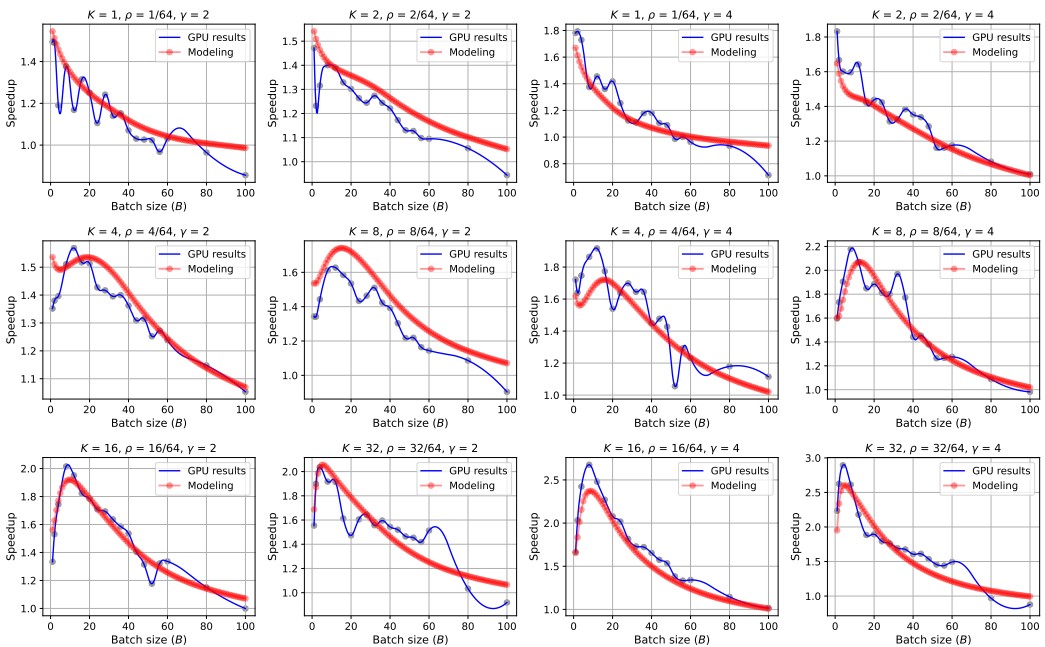

Figure 25: Comparison between GPU results and modeling with 57 measurements.

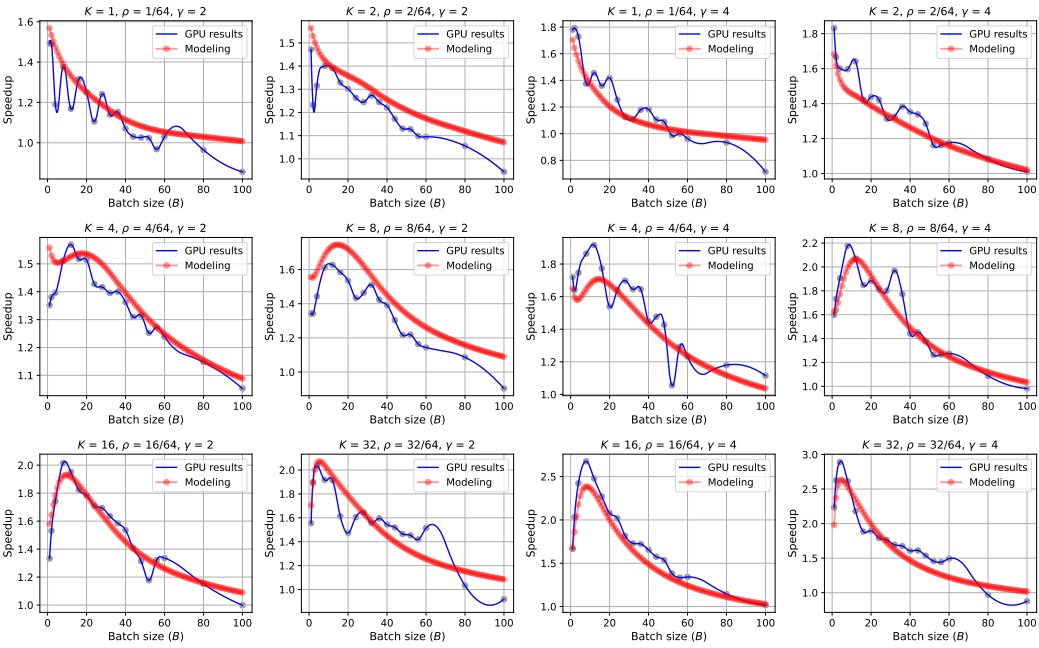

Figure 26: Comparison between GPU results and modeling with 76 measurements.

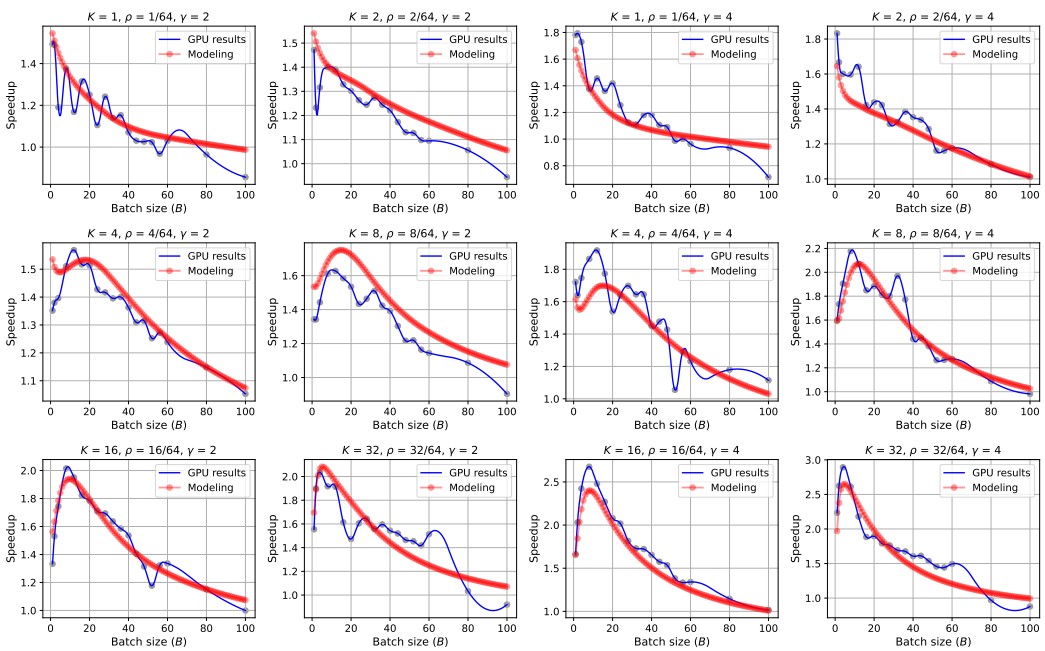

Figure 27: Comparison between GPU results and modeling with 114 measurements.

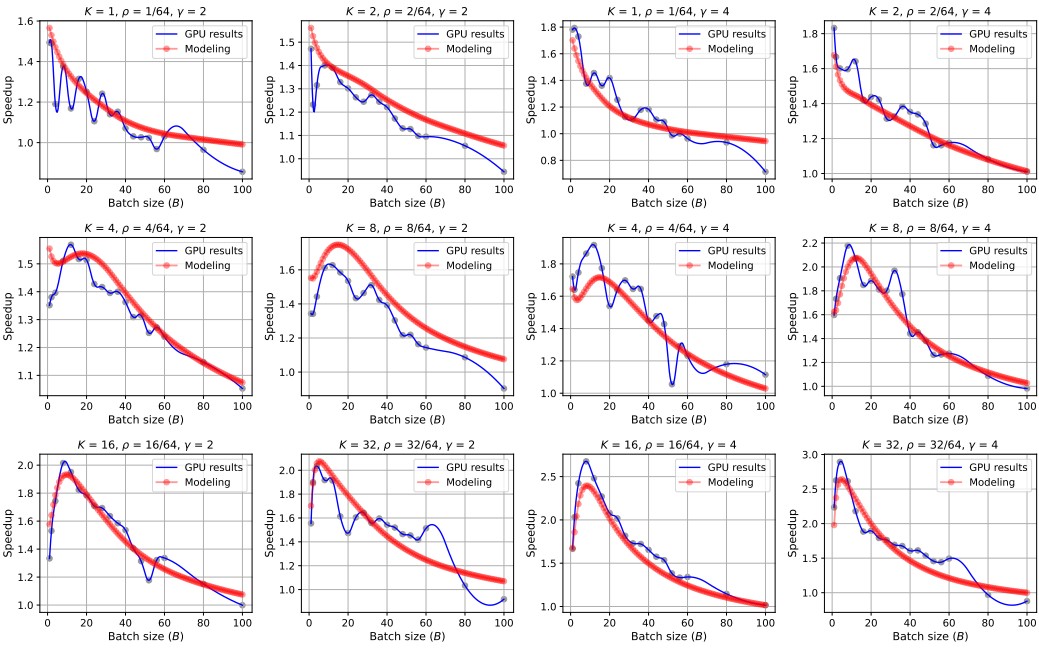

Figure 28: Comparison between GPU results and modeling with 228 measurements.

