# OpenReview forum: "MoESD: Unveil Speculative Decoding's Potential for Accelerating Sparse MoE"
_NeurIPS.cc/2025/Conference — NeurIPS 2025 spotlight_

### Official Review · Reviewer_Xih7 · 2025-06-04

**Clarity:** 3
**Significance:** 3
**Originality:** 3
**Rating:** 4
**Confidence:** 1

**Summary:**

This paper revisits speculative decoding (SD) in the context of Mixture-of-Experts (MoE) models and challenges the widely held belief that SD is ineffective for sparse MoEs due to verification overhead. The authors introduce a theoretical and empirical framework that shows under moderate batch sizes, SD can offer greater acceleration for sparse MoEs than dense models. A new metric, target efficiency, is proposed to analyze SD speedup potential independent of the acceptance rate. Experimental results on Qwen2 models and two benchmarks (HumanEval and MT-bench) validate the theory and show up to 2.29× speedup.

**Questions:**

1. Have the authors considered hybrid metrics that combine target efficiency and acceptance rate to better predict speedup?
2. Beyond moderate batch sizes, SD performance degrades. Are there strategies (e.g., chunked batch processing) that can extend SD benefits in such regimes?
3. The paper uses a fixed draft model. Could training a task-specific or sparsity-aware draft model further improve the speedup or acceptance rate?

**Ethical Concerns:**

["NO or VERY MINOR ethics concerns only"]

**Final Justification:**

I think the reviewers answered my questions well. I have no further questions to this submission and will maintain my current score.

**Limitations:**

Yes.

**Paper Formatting Concerns:**

No.

**Quality:**

3

**Strengths And Weaknesses:**

Strengths:
1. The core claim that SD can outperform on sparse MoEs under moderate batch sizes is counterintuitive and well-motivated.
2. The formulation of the target efficiency metric adds analytical clarity

Weaknesses:
1. While Qwen2 is used with MT-bench and HumanEval, additional LLMs (e.g., Llama, Mistral) would strengthen generalization claims.

---

> ### Author Rebuttal · Authors · 2025-07-30
>
> Thank you very much for your careful review and constructive feedback! Your insightful questions have provided valuable directions for future improvements and greatly inspired our thinking. Our responses are as follows, and we hope they can resolve your questions and concerns.
>
> # Weakness: Model diversity.
>
> We have added experiments for another MoE: Mixtral-8x7B-Instruct-v0.1. We used Eagle's **speculation head** for Mixtral rather than a standalone draft model to validate our method's generality. Due to the format limitations of this year's rebuttal, we can only present the results in tables as follows. Your advice on model diversity is very valuable and we will add them to the revised paper.
>
> As shown in tables, experimental results on another MoE also **conform to our theoretical analysis**, where the speculative decoding (SD) speedups on MoE models first increase and then decrease as batch size increases. Since our theoretical analysis places no restrictions on model architectures, this result is as expected and should hold for other MoE models.
>
> **Table 1: The SD speedup trend vs. batch size, under different settings for Mixtral-8x7B-Instruct-v0.1**
>
> |  Batch sizes   | 2    | 4    | 6    | 8    | 10   | 12   | 16   | 20   | 24   | 28   | 32   | 36   | 40   | 60   | 80   | 100  |
> |----|-|-|-|-|-|-|-|-|-|-|-|-|-|-|-|-|
> | Speedup @(humaneval, temperature 0) | 0.90 | 0.96 | 1.19 | 1.04 | 1.71 | 1.67 | 1.43 | 1.31 | 1.21 | 1.18 | 1.14 | 1.11 | 1.15 | 0.97 | 0.80 | 0.72 |
> | Speedup @(humaneval, temperature 1) | 0.91 | 0.91 | 1.07 | 0.99 | 1.37 | 1.38 | 1.16 | 1.13 | 1.01 | 1.06 | 1.07 | 1.05 | 1.04 | 0.88 | 0.71 | 0.54 |
> | Speedup @(mtbench, temperature 0) | 0.84 | 0.86 | 1.00 | 0.98 | 1.34 | 1.28 | 1.23 | 1.22 | 1.14 | 1.12 | 1.07 | 1.06 | 1.04 | 0.95 | 0.88 | 0.81 |
> | Speedup @(mtbench, temperature 1) | 0.85 | 0.80 | 0.96 | 0.93 | 1.15 | 1.17 | 1.13 | 1.14 | 1.09 | 1.05 | 0.99 | 0.97 | 0.92 | 0.86 | 0.75 | 0.69 |
>
>
> **Table 2: The overview of SD speedup on Mixtral-8x7B-Instruct-v0.1**
>
> *Notation: This table follows the format of Table 1 in the original paper. $\gamma$ represents draft token length, $T_{AR}$ denotes auto-regressive decoding time, $T_{SD}$ denotes speculative decoding time, $\sigma$ is the ratio of actually generated tokens to theoretical maximum, and $\mathbf{x}$ denotes maximal speedup across batch sizes.*
>
> | Dataset   | Temperature | $\gamma=2$  |        |        |        | $\gamma=3$  |        |        |        | $\gamma=4$  |        |        |        |
> |-|---|--|-----|-----|-----|--|-----|-----|-----|--|-----|-----|-----|
> |    |      | $T_{AR}$    | $T_{SD}$      | $\sigma$      | $\mathbf{x}$  | $T_{AR}$    | $T_{SD}$      | $\sigma$      | $\mathbf{x}$  | $T_{AR}$    | $T_{SD}$      | $\sigma$      | $\mathbf{x}$  |
> | humaneval | 0.0  | 20.86       | 12.47  | 0.78   | 1.67   | 21.00       | 12.46  | 0.66   | 1.69   | 20.86       | 11.69  | 0.58   | **1.79**   |
> | humaneval | 1.0  | 21.52       | 15.58  | 0.61   | **1.38**   | 21.39       | 16.03  | 0.46   | 1.33   | 21.48       | 16.23  | 0.39   | 1.32   |
> | mtbench   | 0.0  | 21.61       | 16.10  | 0.61   | **1.34**   | 21.61       | 16.43  | 0.46   | 1.32   | 21.36       | 16.89  | 0.39   | 1.26   |
> | mtbench   | 1.0  | 21.33       | 17.70  | 0.53   | **1.21**   | 21.33       | 17.84  | 0.43   | 1.20   | 21.33       | 18.05  | 0.35   | 1.18   |
>
>
> # Question 1: Consideration of a hybrid metric that combines target efficiency and acceptance rate
>
> A hybrid metric combining target efficiency and acceptance rate would closely approximate the speedup metric itself. As demonstrated in the theoretical derivation in Section 3.1:
>
> $$Speedup = \frac{1-\alpha^{\gamma+1}}{1-\alpha} \cdot \frac{1}{\gamma \cdot \frac{T_D(B,1)}{T_T(B,1)} + \frac{T_T(B,\gamma)}{T_T(B,1)} + \frac{T_{reject}}{T_T(B,1)}}$$
>
> Where $\alpha$ represents the acceptance rate, $\frac{T_T(B,1)}{T_T(B,\gamma)}$ corresponds to our target efficiency metric, $\gamma$ is the average draft token length (a constant). Both $T_D(B,1)$ (draft model inference time) and $T_{reject}$ (rejection sampling overhead) are significantly smaller than $T_T(B,1)$, contributing minimally compared to $\frac{T_T(B,1)}{T_T(B,\gamma)}$.
>
> Therefore, the hybrid metric you suggested would be nearly identical to the speedup formulation above. To avoid redundancy, we chose to directly use the speedup metric rather than introducing an additional combined metric. However, the introduction of target efficiency is critical as acceptance rate alone cannot fully explain the SD speedup trend.
>
>
> # Question 2: Discussion of chunked batch processing
>
> Thank you for this excellent question. Chunked batch processing is indeed a valuable strategy that can help **jointly optimize throughput and latency**.
>
> LLM serving systems must satisfy both throughput requirements and latency constraints such as Time-To-First-Token (TTFT) and Time-Per-Output-Token (TPOT). Generally, there exists a fundamental **trade-off between low latency and high throughput**. When batch sizes are very large, each request receives limited computational resources, potentially failing to meet latency requirements.
>
> For dense models, SD also follows this trade-off [1,2,3]: when speedup is significant, batch sizes are small (limiting throughput); when batch size increases, SD speedup degrades substantially (worse latency). However, our work demonstrates that for SD with MoE models, there actually exists a regime where speedup **increases with** batch sizes. This unique property suggests that chunked batch processing could simultaneously enhance both throughput and latency.
>
> We believe this represents a promising direction for future work, particularly in serving environments where satisfying Service Level Objectives (SLOs) is critical.
>
> # Question 3: Training a task-specific or sparsity-aware draft model to further improve speedup
>
> Regarding task-specific draft models, this is indeed an active area of research. Previous works [4,5,6] have demonstrated that training specialized draft models can improve acceptance rates for specific tasks. Additionally, recent work [7] has explored an online approach that finetunes the draft model with real-time draft token acceptance patterns during serving. Since acceptance rate optimization has been extensively studied and is not our contribution, we chose to use the basic fixed draft model.
>
> Regarding sparsity-aware draft models, we think this approach only has limited benefits. The acceptance rate fundamentally reflects the alignment between draft and target models on a given task. While task-specific training can indeed improve this alignment, sparsity patterns in the target model don't have a direct impact on this. As long as MoE target models achieve similar performance on a task, they are identical in accepting tokens from the draft model's perspective.
>
> # References
>
> [1] Optimizing speculative decoding for serving large language models using goodput.
>
> [2] The synergy of speculative decoding and batching in serving large language models.
>
> [3] Specinfer: Accelerating generative llm serving with speculative inference and token tree verification.
>
> [4] EAGLE: Speculative Sampling Requires Rethinking Feature Uncertainty.
>
> [5] DistillSpec: Improving Speculative Decoding via Knowledge Distillation.
>
> [6] Learning Harmonized Representations for Speculative Sampling.
>
> [7] Online Speculative Decoding.

---

> > ### Comment · Reviewer_Xih7 · 2025-08-05
> >
> > I think the reviewers answered my questions well. I have no further questions to this submission and will maintain my current score.

---

> > > ### Author Response · Authors · 2025-08-08
> > >
> > > We are glad our response answered your questions. We greatly appreciate your thoughtful and highly constructive feedback, and it has been a great pleasure to engage with you in this discussion.

---

### Official Review · Reviewer_WmFj · 2025-07-01

**Clarity:** 3
**Significance:** 2
**Originality:** 3
**Rating:** 4
**Confidence:** 2

**Summary:**

In this paper, the authors challenge the common belief that speculative decoding (SD) is less efficient for MoE models. With both theoretical justification and empirical experiments, the authors show that SD can be more efficient for MoE when the batch size is moderate and MoE is sparse.

For the theoretical part, the authors focus on a commonly negligible factor, target efficiency, instead of the SD optimization, and show that SD's efficiency is more related to this target efficiency instead of acceptance rate, which also makes the acceleration process transparent and explainable.

With practical experiments on a large MoE model (Qwen2-57B-17A), the authors show a good correlation between the observation and theoretical justification. The results also show that SD is less efficient for dense models for moderate batch size.

**Questions:**

I'm open to raise my evaluation score if the authors could properly clarify my 2 weakness points.

**Ethical Concerns:**

["NO or VERY MINOR ethics concerns only"]

**Final Justification:**

In the discussion, the authors well address my misunderstanding of the counter-intuition and batch size. So I increase the score to 4 (botherline accept).

**Limitations:**

yes

**Quality:**

2

**Strengths And Weaknesses:**

**Note**: I'm not very qualified for reviewing this paper, since I'm more an engineering researcher. I focus more on the engineering part.

## Strengths
- The paper is well-written with clear motivation and thorough explanation of the design choice;
- Different from previous works that mostly focus on the algorithm optimization, this paper focuses on a commonly negligible factor when analyzing SD's efficiency, and show a very good correlation between the formula and the observation.

## Weaknesses
- The findings are very intuitive from my perspective. For a small batch size, MoE is inclined to memory-bound. For a large batch size, MoE is inclined to computation-bound. So a moderate batch size will be efficient for MoE.
- The paper shows that SD works well for sparse MoE with a moderate batch size, hinting its limitation.

---

> ### Author Rebuttal · Authors · 2025-07-30
>
> Thank you for your careful reading and thoughtful comments on our paper! We provide our responses below and hope they address your concerns.
>
> # Weakness 1: Clarify the counterintuitive part of our paper's findings.
>
> There seem to be some misunderstandings of our work. We are not arguing that **MoE itself** favours moderate batch sizes, but rather that **SD acceleration for MoE** favours moderate batch sizes.
>
> Given that speculative decoding (SD) leverages spare computational resources caused by memory-boundedness, SD acceleration is expected to be **more pronounced at small batch sizes**, which has been widely observed in previous SD works [1,2,3]. Therefore, your statement that "*For a small batch size, MoE is inclined to memory-bound... So a moderate batch size will be efficient for MoE*" is incorrect — SD should be more effective under memory-bound conditions.
>
> The **counterintuitive** part for MoE is that SD is ineffective at small batch sizes, but becomes **more effective as batch size increases** to moderate level. This is because at small batch sizes, MoE's multi-token verification becomes significantly slower than single-token decoding, leading to degraded target efficiency (the metric we have proposed in paper). Computing multiple tokens often activates more experts, resulting in increased parameter loading and therefore prolonging execution time.
>
> Beyond the *preliminary* conclusion of SD speedup trend on MoE mentioned above, we *further* investigate the impact of MoE sparsity on SD speedup. Through theoretical analysis and experiments, we show that SD is actually **more effective for sparser MoE models**. The table below compares SD speedup between MoE models and dense models (which can be viewed as completely non-sparse MoE models) under identical settings. While MoE initially shows inferior acceleration ratios, it subsequently achieves more obvious acceleration over an extended range.
>
> **Table: Comparison of MoE speedup on the MoE model (Qwen2-57B-A14B-Instruct) and the dense model (OPT),  corresponding to Figure 2c in the original paper.**
> | batch size | 1 | 2 | 4 | 8 | 12 | 16 | 20 | 24 | 28 | 32 | 36 | 40 | 44 | 48 | 52 | 56 | 60 | 80 |
> | - | - | - | - | - | - | - | - | - | - | - | - | - | - | - | - | - | - | - |
> | SD speedup for dense model | 1.807 | 1.418 | 1.346 | 1.261 | 1.177 | 1.085 | 1.038 | 0.947 | 0.848 | 0.839 | 0.797 | 0.756 | 0.712 | 0.698 | 0.665 | 0.625 | 0.648 | 0.585 |
> | SD speedup for moe model | 1.578 | 1.381 | 1.152 | 1.511 | 1.717 | 1.414 | 1.437 | 1.397 | 1.398 | 1.281 | 1.179 | 1.167 | 1.156 | 1.114 | 1.072 | 1.059 | 1.012 | 0.877 |
> | $\frac{\text{SD speedup for moe model}}{\text{SD speedup for dense model}}$ | 0.874 | 0.974 | 0.856 | **1.198** | **1.459** | **1.303** | **1.385** | **1.475** | **1.648** | **1.527** | **1.479** | **1.543** | **1.624** | **1.596** | **1.612** | **1.695** | **1.563** | **1.498** |
>
>
> The beliefs that SD is unsuitable for MoE and degrades with increasing batch sizes have been long-held in the community[1,2,3,4,5]. Consequently, SD research has focused primarily on dense models for edge deployment, overlooking the potential for recently popular MoE architectures. Our work breaks this conventional wisdom by identifying opportunities for SD in other regimes, thereby helping the community develop a more comprehensive understanding of SD.
>
> In summary, our paper effectively challenges traditional viewpoints and provides **counterintuitive** findings as follows:
> * SD is more suitable for dense models than MoEs [4,5] → We demonstrate that at moderate batch sizes, SD is more effective for sparse MoE models.
> * SD effectiveness decreases with increasing batch size [1,2,3] → We demonstrate that for MoE, effectiveness may actually increase with growing batch size.
>
>
>
> # Weakness 2: Concerns about moderate batch size
>
> We would like to clarify the practical value of our work from both application and model perspectives.
>
> **From the application perspective**
>
> * Moderate batch sizes are common in private serving scenarios, which are becoming increasingly popular due to their ability to ensure data security. For example, major AI companies now provide internal *enterprise chatbot* services for their employees.
>
> * When latency requirements are strict, large batch sizes are often not feasible. LLM serving must satisfy not only throughput requirements but also other service level objectives (SLOs) including time-to-first-token (TTFT) and time-per-output-token (TPOT). Excessively large batches limit the computational resources allocated to each request, leading to latency violations. In such cases, moderate batch sizes are necessary.
>
> * From a high-level perspective, our work reveals that applying SD on MoE can **relax the trade-off between latency and throughput**. In most cases, latency and throughput cannot be optimized simultaneously. For example, when applying SD on dense models, speedup diminishes as batch size increases. However, we have shown that for MoE models, there exists a regime where SD speedup increases (lower latency) with larger batch sizes (larger throughput).
>
> **From the model perspective**
>
> Compared to dense models, MoE models are actually *least efficient* in moderate batch sizes. At small batch sizes, MoE models use a small portion of FFN parameters. At large batch sizes, all loaded parameters are provided with abundant computation. However, at moderate batch sizes, all experts are activated while the workload provides insufficient computation, leading to underutilized computational resources. Therefore, optimizing MoE in this regime is critical. Our work demonstrates that SD is a viable solution for such an efficiency gap.
>
>
> # Reference
>
> [1] Optimizing speculative decoding for serving large language models using goodput.
>
> [2] The synergy of speculative decoding and batching in serving large language models.
>
> [3] Specinfer: Accelerating generative llm serving with speculative inference and token tree verification.
>
> [4] EAGLE: Speculative Sampling Requires Rethinking Feature Uncertainty.
>
> [5] Utility-Driven Speculative Decoding for Mixture-of-Experts.

---

> > ### Comment · Reviewer_WmFj · 2025-08-06
> >
> > Thank you very much for your detailed clarification that well clarify my doubt. I will increase the score.

---

> > > ### Author Response · Authors · 2025-08-08
> > >
> > > Thank you for your careful review of our rebuttal. We are glad our response addressed your concerns and greatly appreciate your decision to raise the score. Your highly constructive feedbacks help to improve the quality of our paper.

---

### Official Review · Reviewer_kQQx · 2025-07-02

**Clarity:** 3
**Significance:** 3
**Originality:** 3
**Rating:** 5
**Confidence:** 4

**Summary:**

This paper studies speculative decoding (SD) for MoE-based LLMs from a new perspective: identifying the conditions under which MoE LLMs benefit most from SD. Contrary to the common assumption that SD may not accelerate MoE-based LLMs, the paper shows that SD can be effective at moderate batch sizes, where experts are typically fully utilized. The benefits become larger when the LLMs become sparser. To support the theoretical analysis, the paper introduces a new metric (i.e. target efficiency) and a modeling approach to formulate optimal SD configurations. Experimental results confirm that the theoretical predictions align closely with real-world performance across different GPUs (e.g., 2×A100, 2×H100) and tasks (e.g., HumanEval, MT-Bench).

**Questions:**

N.A.

**Ethical Concerns:**

["NO or VERY MINOR ethics concerns only"]

**Final Justification:**

The reviewer appreciated the authors' efforts on extra experiments. The reviewer'd like to keep the postive rating of the paper.

**Limitations:**

Limitations (e.g. longer context/KV caches) have been discussed in the paper.

**Paper Formatting Concerns:**

N.A.

**Quality:**

3

**Strengths And Weaknesses:**

---
Strengths

i) Topic: This paper explores SD for MoE and general LLMs from a different perspective—identifying the conditions under which SD provides the most benefit. The topic is timely and interesting, addressing an important gap in current research.

ii) Methodology: To support the theoretical analysis, the paper introduces a new metric (i.e. target efficiency) and a modeling approach to derive optimal SD configurations. The formulation is well-motivated and technically sound.

iii) Experiments: Empirical results demonstrate close alignment between the theoretical predictions and real-world performance across various GPUs (e.g., 2×A800, 2×H800) and benchmarks (e.g., HumanEval, MT-Bench).

---
Weaknesses

The paper would be stronger with more comprehensive evaluations across a broader range of LLMs and hardware configurations.

i) Model diversity: The current experiments are based on a single pair of target and draft models—Qwen2-57B-A17B-Instruct and Qwen2-0.5B-Instruct—which limits the generalizability of the findings. The authors are encouraged to include additional representative model pairs, ideally with varying sizes and architectures.

ii) Hardware settings: All experiments were conducted on 2xA800 or 2xH800 GPUs. While informative, these settings may not fully reflect real-world deployment scenarios, which often involve distributed environments with more inter-GPU communication or different compute-memory tradeoffs (e.g., H800 vs. L20).

---

> ### Author Rebuttal · Authors · 2025-07-30
>
> Thank you for your valuable suggestions on diversity, which help to make our conclusion more convincing. We have supplemented experiments, hoping they can address your concerns.
>
> # Weakness 1:  Model diversity
>
> We have added experiments for another MoE: Mixtral-8x7B-Instruct-v0.1. We used Eagle's **speculation head** for Mixtral rather than a standalone draft model to validate our method's generality. Due to the format limitations of this year's rebuttal, we can only present the results in tables as follows. Your advice on model diversity is very valuable and we will add them to the revised paper.
>
> As shown in tables, experimental results on another MoE also **conform to our theoretical analysis**, where the speculative decoding (SD) speedups on MoE models first increase and then decrease as batch size increases. Since our theoretical analysis places no restrictions on model architectures, this result is as expected and should hold for other MoE models.
>
> **Table 1: The SD speedup trend vs. batch size, under different settings for Mixtral-8x7B-Instruct-v0.1**
>
> |  Batch sizes   | 2    | 4    | 6    | 8    | 10   | 12   | 16   | 20   | 24   | 28   | 32   | 36   | 40   | 60   | 80   | 100  |
> |----|-|-|-|-|-|-|-|-|-|-|-|-|-|-|-|-|
> | Speedup @(humaneval, temperature 0) | 0.90 | 0.96 | 1.19 | 1.04 | 1.71 | 1.67 | 1.43 | 1.31 | 1.21 | 1.18 | 1.14 | 1.11 | 1.15 | 0.97 | 0.80 | 0.72 |
> | Speedup @(humaneval, temperature 1) | 0.91 | 0.91 | 1.07 | 0.99 | 1.37 | 1.38 | 1.16 | 1.13 | 1.01 | 1.06 | 1.07 | 1.05 | 1.04 | 0.88 | 0.71 | 0.54 |
> | Speedup @(mtbench, temperature 0) | 0.84 | 0.86 | 1.00 | 0.98 | 1.34 | 1.28 | 1.23 | 1.22 | 1.14 | 1.12 | 1.07 | 1.06 | 1.04 | 0.95 | 0.88 | 0.81 |
> | Speedup @(mtbench, temperature 1) | 0.85 | 0.80 | 0.96 | 0.93 | 1.15 | 1.17 | 1.13 | 1.14 | 1.09 | 1.05 | 0.99 | 0.97 | 0.92 | 0.86 | 0.75 | 0.69 |
>
>
> **Table 2: The overview of SD speedup on Mixtral-8x7B-Instruct-v0.1**
>
> *Notation: This table follows the format of Table 1 in the original paper. $\gamma$ represents draft token length, $T_{AR}$ denotes auto-regressive decoding time, $T_{SD}$ denotes speculative decoding time, $\sigma$ is the ratio of actually generated tokens to theoretical maximum, and $\mathbf{x}$ denotes maximal speedup across batch sizes.*
>
> | Dataset   | Temperature | $\gamma=2$  |        |        |        | $\gamma=3$  |        |        |        | $\gamma=4$  |        |        |        |
> |-|---|--|-----|-----|-----|--|-----|-----|-----|--|-----|-----|-----|
> |    |      | $T_{AR}$    | $T_{SD}$      | $\sigma$      | $\mathbf{x}$  | $T_{AR}$    | $T_{SD}$      | $\sigma$      | $\mathbf{x}$  | $T_{AR}$    | $T_{SD}$      | $\sigma$      | $\mathbf{x}$  |
> | humaneval | 0.0  | 20.86       | 12.47  | 0.78   | 1.67   | 21.00       | 12.46  | 0.66   | 1.69   | 20.86       | 11.69  | 0.58   | **1.79**   |
> | humaneval | 1.0  | 21.52       | 15.58  | 0.61   | **1.38**   | 21.39       | 16.03  | 0.46   | 1.33   | 21.48       | 16.23  | 0.39   | 1.32   |
> | mtbench   | 0.0  | 21.61       | 16.10  | 0.61   | **1.34**   | 21.61       | 16.43  | 0.46   | 1.32   | 21.36       | 16.89  | 0.39   | 1.26   |
> | mtbench   | 1.0  | 21.33       | 17.70  | 0.53   | **1.21**   | 21.33       | 17.84  | 0.43   | 1.20   | 21.33       | 18.05  | 0.35   | 1.18   |
>
>
>
> # Weakness 2: Hardware settings diversity
>
>
> We add experiments for additional hardware settings including **more inter-GPU communication** and **different compute-memory tradeoff** as you required. The configurations include: 2×A800 (original setup for comparison), 4×A800 (newly added), and 4×L40 (newly added). Since L40 only has 48GB memory while the Qwen2 MoE model has 57B parameters, we use 4×L40 due to memory constraints.
>
> **More inter-GPU Communication:** 2×A800 (1st group) and 4×A800 (2nd group) configurations in Table 3 are compared. For absolute timings (i.e., $T_{AR}$ and $T_{SD}$), we observe obvious decreases due to additional devices. For SD speedups,  they maintain similar trends but with slight degradation.  This occurs because while the large model is parallelized between GPUs, the small draft model remains on a single GPU, relatively increasing its forward pass time compared to the large model's forward time.
>
> **Different compute-memory tradeoffs:** 4×A800 (2nd group) and 4×L40 (3rd group) configurations in Table 3 are compared. A800-80GB provides 312 TFlops BF16 performance with 1.94 TB/s bandwidth (roofline ridge point ≈161), while L40 offers 181 TFlops with 864 GB/s bandwidth (ridge point ≈214). L40's higher ridge point indicates more available compute capacity for SD verification, enabling higher speedups as confirmed by our experimental results.
>
> **Table 3: SD speedup on different hardware settings**
>
> *Notation: This table follows the format of Table 1 in the original paper. $\gamma$ represents draft token length, $T_{AR}$ denotes auto-regressive decoding time, $T_{SD}$ denotes speculative decoding time, $\sigma$ is the ratio of actually generated tokens to theoretical maximum, and $\mathbf{x}$ denotes maximal speedup across batch sizes.*
>
> | Device  | Dataset  | Temp | $\gamma=2$ |  |  |  | $\gamma=3$ |  |  |  | $\gamma=4$ |  |  |  |
> |----|-----|-|--|---|---|---|--|---|---|---|--|---|---|---|
> | |  |      | $T_{AR}$  | $T_{SD}$ | $\sigma$ | $\mathbf{x}$     | $T_{AR}$  | $T_{SD}$ | $\sigma$ | $\mathbf{x}$     | $T_{AR}$  | $T_{SD}$ | $\sigma$ | $\mathbf{x}$     |
> | 2xA800  | humaneval| 0.0  | 18.89     | 11.61    | 0.94     | 1.63     | 15.93     | 8.11     | 0.93     | 1.96     | 15.93     | 7.31     | 0.91     | **2.18** |
> | | humaneval| 1.0  | 19.13     | 12.93    | 0.83     | 1.48     | 21.20     | 14.09    | 0.73     | 1.50     | 19.13     | 11.14    | 0.67     | **1.72** |
> | | mtbench  | 0.0  | 20.92     | 16.70    | 0.71     | 1.25     | 16.00     | 12.43    | 0.62     | **1.29** | 20.92     | 17.53    | 0.55     | 1.19     |
> | | mtbench  | 1.0  | 21.15     | 17.33    | 0.68     | 1.22     | 19.09     | 14.83    | 0.57     | **1.29** | 19.09     | 15.93    | 0.48     | 1.20     |
> | 4xA800  | humaneval| 0.0  | 11.20     | 6.77     | 0.95     | 1.65     | 11.20     | 5.89     | 0.93     | 1.90     | 11.20     | 5.38     | 0.90     | **2.08** |
> | | humaneval| 1.0  | 11.72     | 8.51     | 0.81     | 1.38     | 12.05     | 8.30     | 0.73     | 1.45     | 11.23     | 7.70     | 0.67     | **1.46** |
> | | mtbench  | 0.0  | 11.26     | 8.92     | 0.72     | **1.26**     | 11.26     | 9.10     | 0.61     | 1.24     | 11.26     | 9.82     | 0.52     | 1.15 |
> | | mtbench  | 1.0  | 11.78     | 10.32    | 0.67     | 1.14     | 11.30     | 9.42     | 0.58     | **1.20**     | 11.78     | 11.25    | 0.47     | 1.05 |
> | 4xL40   | humaneval| 0.0  | 17.84     | 10.00    | 0.95     | 1.79     | 17.84     | 8.33     | 0.93     | 2.14     | 17.84     | 7.94     | 0.90     | **2.25** |
> | | humaneval| 1.0  | 17.89     | 12.27    | 0.80     | 1.46     | 17.89     | 11.07    | 0.74     | 1.62     | 17.89     | 10.91    | 0.65     | **1.64** |
> | | mtbench  | 0.0  | 20.40     | 15.87    | 0.71     | **1.29**     | 20.40     | 16.22    | 0.62     | 1.26     | 20.40     | 16.33    | 0.54     | 1.25 |
> | | mtbench  | 1.0  | 20.58     | 16.02    | 0.68     | **1.28**     | 18.11     | 14.75    | 0.54     | 1.23     | 18.11     | 15.54    | 0.48     | 1.17 |
>
> We also present how speedup varies with batch size in Table 4. As shown, the speedup demonstrates a consistent trend that aligns with the theoretical analysis and experiments reported in the original paper: the speedup initially increases and then decreases, achieving peak performance at moderate batch sizes.
>
> **Table 4: SD speedup vs. batch sizes on different hardware settings**
> |  Batch sizes   | 1    | 2    | 4    | 8   | 12   | 16   | 20   | 24   | 28   | 32   | 36   | 40   | 60   | 80   | 100  |
> |----|-|-|-|-|-|-|-|-|-|-|-|-|-|-|-|
> | 2xA800 | 1.60 | 1.73 | 1.91 | 2.18 | 2.07 | 1.85 | 1.88 | 1.81 | 1.80 | 1.97 | 1.77 | 1.44 | 1.27 | 1.09 | 0.98 |
> | 4xA800 | 1.47 | 0.57 | 1.74 | 1.87 | 2.08 | 1.55 | 1.72 | 1.56 | 1.56 | 1.45 | 1.40 | 1.45 | 1.12 | 1.04 | 0.90 |
> | 4xL40| 1.31 | 1.57 | 1.85 | 2.16 | 2.25 | 1.86 | 1.95 | 1.84 | 1.86 | 1.84 | 1.80 | 1.59 | 1.33 | 1.16 | 0.94 |
>
> In summary, the SD speedups in scenarios with more inter-GPU communication or compute-memory tradeoffs **conforms to our theoretical analysis as expected**. We will include these results in the revised paper to strengthen the generalizability of our findings.

---

> > ### Author Response · Authors · 2025-08-08
> >
> > As the discussion phase is approaching its end, we would kindly ask the reviewer to let us know if the above added experiments have addressed the your concerns. We greatly appreciate the reviewer for engaging with us in the discussion.

---

### Official Review · Reviewer_8fAG · 2025-07-02

**Clarity:** 2
**Significance:** 3
**Originality:** 3
**Rating:** 5
**Confidence:** 2

**Summary:**

This paper applies speculative decoding (SD) to sparse Mixture-of-Experts (MoE) models for inference acceleration. The authors demonstrate that the combination yields higher speedups at moderate batch sizes. The paper introduces a new metric called target efficiency for speedup measurment in MoE models. Evaluation results using Qwen2-57B-A17B-Instruct as verifier and Qwen2-0.5B-Instruct as light-weight drafter shows up to 2.29$\times$ speedup.

**Questions:**

- The authors states multiple times that they challenge the conventional belief that speculative decoding cannot effectively accelerate MoEs, where the “belief” is more of an untested assumption than a well-established empirical claim. It would be useful to cite relevant prior work on this.
- To help illustrate the framework, the authors could consider including a system diagram to show the details of MoESD workflow and design?
- Minor point: It would be helpful to walk-through Algo 1 in the text.

**Ethical Concerns:**

["NO or VERY MINOR ethics concerns only"]

**Final Justification:**

My concerns have been sufficiently addressed, and I am raising my score to Accept.

**Limitations:**

Yes

**Quality:**

2

**Strengths And Weaknesses:**

Strengths:

- The paper studies the niche area at the intersection of MoE and SD, which is a timely topic - MoE brings state-of-the-art generation quality and SD is well-established for speedup at inference time.
- The paper introduces a new SD performance metric for MoE setting, namely target efficiency, which goes beyond commonly known acceptance rate by measuring how much verifier work is converted into useful tokens.
- The theoretical modelling seems to be sound.

Weaknesses:

- My main concern is that the evaluation appears to be narrow as only one sparse model was evaluated (although OPT is included for dense model case). Given this, and the standard that NeurIPS holds for submissions, I will have to request
- The evaluation doesn’t seem to include tree-based speculative decoding e.g. Eagle, Eagle-2, which are at the top of the SD leader-board according to SpecBench (https://github.com/hemingkx/Spec-Bench/blob/main/Leaderboard.md). It would hence be useful to include analysis of such tree-based method.

---

> ### Author Rebuttal · Authors · 2025-07-30
>
> Thank you sincerely for your constructive feedback! We hope our responses below can address your concerns and clarify any misunderstandings about our work.
>
> # Weakness 1: Model diversity.
>
> We have added experiments for another MoE: Mixtral-8x7B-Instruct-v0.1. We used Eagle's **speculation head** for Mixtral rather than a standalone draft model to validate our method's generality. Due to the format limitations of this year's rebuttal, we can only present the results in tables as follows. Your advice on model diversity is very valuable and we will add them to the revised paper.
>
> As shown in tables, experimental results on another MoE also **conform to our theoretical analysis**, where the speculative decoding (SD) speedups on MoE models first increase and then decrease as batch size increases. Since our theoretical analysis places no restrictions on model architectures, this result is as expected and should hold for other MoE models.
>
> **Table 1: The SD speedup trend vs. batch size, under different settings for Mixtral-8x7B-Instruct-v0.1**
>
> |  Batch sizes   | 2    | 4    | 6    | 8    | 10   | 12   | 16   | 20   | 24   | 28   | 32   | 36   | 40   | 60   | 80   | 100  |
> |----|-|-|-|-|-|-|-|-|-|-|-|-|-|-|-|-|
> | Speedup @(humaneval, temperature 0) | 0.90 | 0.96 | 1.19 | 1.04 | 1.71 | 1.67 | 1.43 | 1.31 | 1.21 | 1.18 | 1.14 | 1.11 | 1.15 | 0.97 | 0.80 | 0.72 |
> | Speedup @(humaneval, temperature 1) | 0.91 | 0.91 | 1.07 | 0.99 | 1.37 | 1.38 | 1.16 | 1.13 | 1.01 | 1.06 | 1.07 | 1.05 | 1.04 | 0.88 | 0.71 | 0.54 |
> | Speedup @(mtbench, temperature 0) | 0.84 | 0.86 | 1.00 | 0.98 | 1.34 | 1.28 | 1.23 | 1.22 | 1.14 | 1.12 | 1.07 | 1.06 | 1.04 | 0.95 | 0.88 | 0.81 |
> | Speedup @(mtbench, temperature 1) | 0.85 | 0.80 | 0.96 | 0.93 | 1.15 | 1.17 | 1.13 | 1.14 | 1.09 | 1.05 | 0.99 | 0.97 | 0.92 | 0.86 | 0.75 | 0.69 |
>
>
> **Table 2: The overview of SD speedup on Mixtral-8x7B-Instruct-v0.1**
>
> *Notation: This table follows the format of Table 1 in the original paper. $\gamma$ represents draft token length, $T_{AR}$ denotes auto-regressive decoding time, $T_{SD}$ denotes speculative decoding time, $\sigma$ is the ratio of actually generated tokens to theoretical maximum, and $\mathbf{x}$ denotes maximal speedup across batch sizes.*
>
> | Dataset   | Temperature | $\gamma=2$  |        |        |        | $\gamma=3$  |        |        |        | $\gamma=4$  |        |        |        |
> |-|---|--|-----|-----|-----|--|-----|-----|-----|--|-----|-----|-----|
> |    |      | $T_{AR}$    | $T_{SD}$      | $\sigma$      | $\mathbf{x}$  | $T_{AR}$    | $T_{SD}$      | $\sigma$      | $\mathbf{x}$  | $T_{AR}$    | $T_{SD}$      | $\sigma$      | $\mathbf{x}$  |
> | humaneval | 0.0  | 20.86       | 12.47  | 0.78   | 1.67   | 21.00       | 12.46  | 0.66   | 1.69   | 20.86       | 11.69  | 0.58   | **1.79**   |
> | humaneval | 1.0  | 21.52       | 15.58  | 0.61   | **1.38**   | 21.39       | 16.03  | 0.46   | 1.33   | 21.48       | 16.23  | 0.39   | 1.32   |
> | mtbench   | 0.0  | 21.61       | 16.10  | 0.61   | **1.34**   | 21.61       | 16.43  | 0.46   | 1.32   | 21.36       | 16.89  | 0.39   | 1.26   |
> | mtbench   | 1.0  | 21.33       | 17.70  | 0.53   | **1.21**   | 21.33       | 17.84  | 0.43   | 1.20   | 21.33       | 18.05  | 0.35   | 1.18   |
>
>
> # Weakness 2: Exclusion of tree-based speculative decoding.
>
> We would like to point out that while tree-based speculation can achieve higher speedup at batch size = 1, they perform poorly in **batched inference**. Compared to chain-based methods, although tree-based methods accept more tokens in each iteration, they have significantly lower values of $\frac{\text{accepted tokens count}}{\text{total draft tokens count}}$, since only one path in the tree can be accepted while all other branches are discarded. Specifically:
> * At batch size = 1, there are abundant idle computational resources, so verifying a large number of draft tokens does not incur significant latency overhead, making tree-based speculation schemes suitable.
> * As batch size increases, there are fewer idle computational resources, and the large number of draft tokens slows down the verification stage. Moreover, most of the verified tokens are destined to be rejected, which wastes computational power.
>
> Additionally, tree-based methods are not engineering-friendly, as they require separate handling of attention masks for each request in the batch. Other works involving **batched** speculative decoding such as [1,2], also adopt chain-based approaches.
>
> # Question 1: Is the belief that speculative decoding cannot effectively accelerate MoEs an untested assumption or a well-established empirical claim?
>
> This belief is supported by **experimental results rather than untested assumptions**. We have *already* provided a citation supporting this belief in line 36 of the original paper, which refers to EAGLE [3]. Table 2 and Table 3 of EAGLE shows that SD achieves 2.54× better speedup on dense models than MoE models. Additionally, [4] from Georgia Tech and Nvidia Research also explicitly states in their abstract that "speculation is ineffective for MoEs" and reports 1.5× SD degradation when applied to MoE. We appreciate the reviewer's constructive feedback, and will incorporate these supporting data in the revised version.
>
> Beyond empirical validation, this belief has solid theoretical grounding. Under small batch sizes, systems are memory-bound, meaning execution time is primarily determined by parameter access. For dense models, decoding one token or verifying multiple tokens loads the same amount of parameters, resulting in similar total execution time. However, for MoE models, verifying multiple tokens means activating more experts and loading additional parameters, thereby increasing latency and reducing speculative decoding efficiency. More rigorous analysis can be found in Section 3.1 in the original paper. This theoretical limitation explains why most existing speculative decoding works focus on accelerating dense models.
>
> # Question 2: Illustration of the system.
>
> Thank you for your constructive suggestions on improving the clarity of our work. We will include a system diagram and a detailed textual walk-through of Algorithm 1 in the revised manuscript.
>
> As figures are not allowed in this year's rebuttal, we provide a textual description of the MoESD framework here. Based on our analysis of roofline effects and the number of activated experts in MoE, we establish the expression of SD speedup for MoE as shown in line 11 of Algorithm 1. Due to uncertainties in operator implementation and GPU architecture, the expression includes several parameters with clear physical meanings for relaxation. Through profiling a small number of speedup data points and lightweight fitting (as detailed in Appendix B), we can determine the values of these parameters. Applying these parameters to the expression yields a complete speedup model. Our experimental results show that this modeling consistently matches real measurements, thereby validating the correctness of our theoretical analysis. This modeling approach also makes the tradeoffs involved in SD acceleration for MoE more transparent and explainable.
>
> # References
>
> [1] MagicDec: Breaking Throughput-Latency Trade-off for Long Context Generation with Speculative Decoding.
>
> [2] BASS: Batched Attention-optimized Speculative Sampling.
>
> [3] EAGLE: Speculative Sampling Requires Rethinking Feature Uncertainty.
>
> [4] Utility-Driven Speculative Decoding for Mixture-of-Experts.

---

> > ### Comment · Reviewer_8fAG · 2025-08-05
> >
> > Dear authors of MoESD,
> >
> > Thank you for your response to my review comments. My concerns have been sufficiently addressed, and I am raising my score to Accept.
> >
> > Best of luck!

---

> > > ### Author Response · Authors · 2025-08-08
> > >
> > > Thank you for your careful review of our rebuttal. We are glad our response addressed your concerns and greatly appreciate your decision to raise the score. Your highly constructive feedbacks help to improve the quality of our paper.

---

### Official Review · Reviewer_jdmT · 2025-07-05

**Clarity:** 3
**Significance:** 2
**Originality:** 2
**Rating:** 4
**Confidence:** 3

**Summary:**

This paper investigates the performance of speculative decoding for Mixture-of-Experts (MoE) models. Specifically, the paper shows that speculative decoding is not as helpful for sparse MoE models as it is for dense models for small batch sizes, when the batch size grows, the benefit substantially increases (which is counter to the trend for dense models). To support these claims, the paper provides theoretical analysis that shows that at moderate batch sizes, the number of activated experts saturates, which helps amortize the additional expert loading costs (in terms of memory bandwidth), and allows for more compute-bound inference. The paper also introduces a new intuitive metric -- target efficiency -- that measures the cost ratio for computing logits for 1 token vs. $\gamma$ tokens (in previous speculative decoding work, this is assumed to be near 1, which is not true for MoE models). Finally, the paper empirically shows the benefits of larger batch sizes on speculative decoding for MoEs with real models.

**Questions:**

- Is there any way to predict what the optimal batch size might be based on your theory, vs through hyper-parameter tuning?
- The comparison to dense models in Figure 2c is insightful but indirect, as it only shows target efficiency.  To make your claim that MoEs can benefit more from SD than dense models more concrete, could you provide the end-to-end speedup graph for the dense model, analogous to Figures 2a and 2b?
- The performance curves in your figures show noticeable fluctuations. While you mention averaging results over five runs, can you also add error bars to Figures 2 and 3?
- Given that MoEs require larger batch sizes, what is the effect on latency, vs throughput?

**Ethical Concerns:**

["NO or VERY MINOR ethics concerns only"]

**Final Justification:**

The paper is well done and the authors adequately addressed my concerns. However, I still believe the novelty is fairly limited, so I am recommending a 4 instead of a 5.

**Quality:**

3

**Strengths And Weaknesses:**

The paper is clearly written and presents a simple finding: sparse MoE models benefit more from speculative decoding under moderate batch sizes. The idea that the acceptance rate is not the only important measurement --- target efficiency is also very important, especially when the assumptions that it is near 1 is violated --- is also a natural additional metric to look at. This could potentially be a valuable finding for practitioners, if it was not already known.

That said, while the paper is well-executed, it is not particularly novel. The core finding, that speculative decoding benefits from instances of memory-bound workloads in MoE models, can be seen as a logical extension of established performance principles. The theoretical analysis, while effective, supports this by applying existing concepts like the roofline model and probability theory, rather than providing new theoretical constructs. There are also some minor gaps in the experimental presentation; the comparison to dense models is indirect, using the "target efficiency" metric instead of a direct end-to-end speedup analysis, and the plots lack error bars, making it somewhat difficult to assess the statistical significance of the results.

---

> ### Author Rebuttal · Authors · 2025-07-30
>
> We sincerely appreciate your careful reading and valuable suggestions for improving our manuscript. In the following response, we first answer your four questions with additional experimental results as supporting evidence, and then respond to the weaknesses you mentioned. We hope these revisions adequately address your concerns.
>
> # Question 1: Is there any way to predict what the optimal batch size might be based on your theory, vs through hyper-parameter tuning?
>
> Predicting speedup based purely on theoretical analysis without using any profiling data is challenging. On GPUs, GEMM is indeed predictable because it involves regular and highly optimized operators. However, this becomes difficult for operators like Attention that involve customized kernel optimizations (FlashAttention1/2, eager attention, SDPA attention), fusion (of different types of nonlinear layers or positional encodings like RoPE and its variants), and multi-hierarchy memory access (registers, caches, DRAM), as pointed out by previous works [1,2,3]. Consequently, other system optimization works like NanoFlow also adopt a two-stage strategy: profiling first, then runtime execution.
>
> We take profiling results of Qwen2-57B-A14B (hidden size 3584) and Mixtral-8x7B (hidden size 4096) as an example to illustrate this. For FFN, Qwen takes a shorter time than Mixtral (143μs vs 226μs), aligning with their relative hidden sizes. However, for Attention, Qwen takes a longer time than Mixtral (271μs vs 115μs), contradicting the theoretical expectation based on hidden size scaling.
>
> Therefore, modeling speedup trends with pure analytical methods requires case-by-case analysis for different operator implementations, GPU microarchitectures, and instruction sets. Moreover, many hardware details are not disclosed by GPU vendors, making such predictions even less practical. In contrast, the hyperparameter approach offers a more general paradigm and is easy to use: all parameters have clear physical meanings, only a few profiling data points are needed, and the computational overhead is very low. Our method achieves a balance between effectiveness and practicality: on one hand, it captures the primary factors (i.e. the number of activated experts and roofline trends); on the other hand, it avoids getting lost in system implementation details.
>
> # Question 2: End-to-end speculative decoding speedup comparsion between MoE and dense models.
>
> We compare target efficiency in Figure 2c in order to isolate the effect of acceptance rate. Given that the strong correlation between target efficiency and speedup has been demonstrated in Figures 2a and 2b, we believe this comparison provides a reliable assessment.
>
> We also have end-to-end (E2E) speedup comparisons, as shown in the table below. As batch sizes increase (>= 16), speculative decoding (SD) indeed achieves greater speedup on MoE models compared to dense models (i.e., the last row exceeds 1). We appreciate the reviewer's constructive feedback, which helps to make our statements more direct and concrete. We will attach these E2E speedup results in the revised paper.
>
> **Table 1 Comparison of E2E SD speedup between MoE and dense models (corresponding to Figure 2c)**
> | batch size|1|2|4|8|12| 16| 20| 24| 28| 32| 36| 40| 44| 48| 52| 56| 60| 80|
> |--|-|-|-|-|-|-|-|-|-|-|-|-|-|-|-|-|-|-|
> | SD speedup for dense model|3.16|2.84|2.57|2.30|2.13|1.83|1.69|1.56|1.44|1.37|1.35|1.23|1.19|1.14|1.04|1.02|1.06|0.96 |
> | SD speedup for moe model|1.60|1.73|1.91|2.18|2.07|1.85|1.88|1.81|1.80|1.97|1.77|1.44|1.45|1.38|1.26|1.27|1.28|1.09 |
> | $\frac{\text{SD speedup for moe model}}{\text{SD speedup for dense model}}$|0.51|0.61|0.74|0.95|0.97|**1.01**|**1.12**|**1.16**|**1.25**|**1.44**|**1.32**|**1.18**|**1.22**|**1.21**|**1.21**|**1.25**|**1.20**|**1.13** |
>
>
> # Question 3: Addition of error bars.
>
> The table below presents the speedup of five independent runs and their average for Figure 2a. The variance in speedup is minimal, confirming the **statistical significance** of our findings. Theoretically, the speedup should also be stable since we fixed the random seed to ensure identical workloads across all runs. Your advice is very helpful, and we will add complete error bars in the revised paper.
>
> **Table 2 Speedup of five independent runs and their average**
> | batch size|1|2|4|8|12| 16| 20| 24| 28| 32| 36| 40| 44| 48| 52| 56| 60| 80| 100  |
> |--|-|-|-|-|-|-|-|-|-|-|-|-|-|-|-|-|-|-|-|
> | run1 speedup|1.61|1.73|1.91|2.20|2.03|1.89|1.94|1.80|1.77|1.91|1.78|1.43|1.40|1.43|1.33|1.20|1.24|1.07|0.90 |
> | run2 speedup|1.56|1.74|1.90|2.19|2.09|1.86|1.86|1.83|1.83|2.00|1.74|1.43|1.43|1.39|1.23|1.26|1.26|1.07|1.02 |
> | run3 speedup|1.61|1.73|1.90|2.20|2.07|1.87|1.79|1.91|1.82|2.00|1.81|1.46|1.50|1.31|1.23|1.32|1.27|1.07|1.00 |
> | run4 speedup|1.61|1.74|1.90|2.16|2.13|1.82|1.86|1.74|1.82|1.95|1.79|1.43|1.48|1.34|1.24|1.27|1.32|1.08|1.02 |
> | run5 speedup|1.59|1.73|1.92|2.15|2.04|1.82|1.98|1.79|1.77|2.00|1.75|1.46|1.47|1.44|1.30|1.29|1.32|1.17|0.98 |
> | speedup std |0.019|0.005|0.006|0.020|0.036|0.029|0.064|0.053|0.025|0.035|0.026|0.013|0.038|0.049|0.042|0.040|0.028|0.038|0.042 |
>
> Regarding the fluctuations in performance curve, they are attributed to GPU's *dimension quantization effects*: tensor core utilization changes with matrix dimensions. When dimensions are not divisible by the GPU's native tile sizes, computation performance degrades. This phenomenon is well-documented in NVIDIA's document [4]. Auto-regressive (AR) decoding is more sensitive to this effect, making the time ratio of AR to SD (namely, SD speedup) fluctuate. Despite these local fluctuations, the overall speedup trend follows our theoretical analysis, confirming the validity of our conclusions.
>
> # Question 4: Latency and throughput tradeoff.
>
> This question reveals a core difference between MoE and dense models when applying SD. Our findings actually demonstrate that for MoE models, latency and throughput can be jointly optimized, while this is hard for dense models.
>
> For dense models, there exists **a trade-off between low latency and high throughput**: smaller batch sizes yield higher SD speedups (low latency but limited throughput), while larger batch sizes result in lower SD speedup (increased throughput but high latency). However, for MoEs, speedup can **increase with** batch size, indicating both latency and throughput are improved. Our work reveals the underlying cause of this *counter-intuitive* phenomenon and further extends it to MoEs with different sparsities.
>
> Your insightful question provides a valuable perspective that helps readers understand our work from a higher level. We will incorporate this into our revised paper.
>
> # Clarification of the novelty and impact of our work.
>
>
> While our work indeed builds upon established principles, we would like to respectfully highlight several aspects that demonstrate novelty and impact.
>
> **Correcting Critical Misconceptions with Quantitative Methodology:** Our work challenges two prevalent beliefs that have constrained SD research as follows:
>
> * SD is more suitable for dense models than MoEs [6,7]. → We demonstrate that at moderate batch sizes, SD is more effective for sparse MoE models.
> * SD performance degrades with larger batch sizes [8,9,10] → We demonstrate that for MoE, effectiveness may actually increase with growing batch size.
>
> Our work is the **first** to systematically refute these beliefs, addressing an important gap in current research. Beyond only *qualitatively* describing SD speedup trends as previous works have done [5,6,10], we further developed a *quantitative* model making SD more explainable, which is also **a first attempt** in SD research. Our theoretical analyses align well with experimental results, providing rigorous support for our conclusions.
>
> **Practical Impact on MoE Optimization:** Beyond theoretical findings, our work has practical impact on MoE optimization. Compared to dense models, MoEs are efficient at both small batches (fewer activated parameters) and large batches (abundant computation), but **suboptimal** at medium batches where full parameters are loaded without sufficient computation to amortize this overhead. Other MoE optimizations such as expert offloading and caching are also ineffective in this regime due to fully activated experts. In this paper, we show that SD—which has been overlooked by previous works for MoE—is effective, thus providing **a novel perspective** on MoE's deficiency at moderate batch sizes.
>
> In summary, though based on existing theories, we have developed **novel conclusions** that help correct misconceptions and optimize MoE models, supported by a quantitative methodology unprecedented in previous SD works. Other reviewers such as kQQx, Xih7 and 8fAG also acknowledge the value of our work on this timely topic of MoE and SD.
>
>
> # Reference
>
> [1] NanoFlow: Towards Optimal Large Language Model Serving Throughput.
>
> [2] Orion: Interference-aware, fine-grained gpu sharing for ml applications.
>
> [3] Dissecting the NVIDIA Volta GPU Architecture via Microbenchmarking.
>
> [4] Since links are not allowed in rebuttal, please search for "Matrix Multiplication Background User's Guide NVIDIA".
>
> [5] MagicDec: Breaking the Latency-Throughput Tradeoff for Long Context Generation with Speculative Decoding
>
> [6] EAGLE: Speculative Sampling Requires Rethinking Feature Uncertainty.
>
> [7] Utility-Driven Speculative Decoding for Mixture-of-Experts.
>
> [8] Optimizing speculative decoding for serving large language models using goodput.
>
> [9] The synergy of speculative decoding and batching in serving large language models.
>
> [10] Specinfer: Accelerating generative llm serving with speculative inference and token tree verification.

---

> > ### Comment · Reviewer_jdmT · 2025-08-07
> >
> > Thank you for the thorough reply to my comments. I have raised my score.

---

> > > ### Author Response · Authors · 2025-08-08
> > >
> > > Thank you for your careful review of our rebuttal. We are glad our response addressed your concerns and greatly appreciate your decision to raise the score. Your highly constructive feedbacks help to improve the quality of our paper.

---

### Decision · Program_Chairs · 2025-09-17

**Decision:**

Accept (spotlight)

**Comment:**

This paper studies whether speculative decoding (SD) is effective for sparse Mixture-of-Experts (MoE) models. It theoretically and empirically demonstrates that MoE models benefit significantly from SD at moderate batch sizes and benefit from SD even more than dense models. The authors introduce a new metric, "target efficiency," to provide a more comprehensive understanding of SD performance beyond the acceptance rate. The paper first introduces some theoretical concepts and then jumps to setting up a framework for establishing a simulation for speedups for target devices/ setups.

**Strengths:**
*   The paper addresses a timely problem and shows that MoEs can benefit more from SD than dense models at moderate batch sizes (kQQx, Xih7).
*   Formalizing the tradeoffs of SD and introducing a "target efficiency", offering a more nuanced way to analyze SD performance beyond simple acceptance rates (jdmT, 8fAG, WmFj).

**Weaknesses:**
*   Initial reviews pointed out a narrow experimental evaluation regarding model diversity (8fAG, kQQx, Xih7) and hardware configurations (kQQx). Some reviewers also noted a lack of direct end-to-end comparisons and statistical error bars in the original submission (jdmT).
* The presentation can be improved a bit to better connect the theory part with the modeling/simulation part. Also, there are many notations that make it hard to follow. The optimization parameters are listed in the appendix which is helpful, it will be good to have a general glossary for all the notations, what they mean, and how to obtain them (if they represent an empirical measure).

Overall the paper provides a valuable analysis with a clear message (SD benefits for MoE) and establishing a simulation framework to benchmark expected gains. The authors provided valuable clarifications and measurements during the rebuttal. Improving the presentation and also linking the code (provided in supplemental) and clear guidelines for obtaining measurements and setting up new benchmarks should provide a good resource for researchers in the field.